# DiSA: Diffusion Step Annealing in Autoregressive Image Generation

## Abstract

An increasing number of autoregressive (AR) models, such as MAR, FlowAR, xAR, and Harmon adopt diffusion sampling to improve the quality of image generation. However, this strategy leads to low inference efficiency, because it usually takes 50 to 100 steps for diffusion to sample a token. This paper explores how to effectively address this issue. Our key motivation is that as more tokens are generated during the AR process, subsequent tokens follow more constrained distributions and are easier to sample. To intuitively explain, if a model has generated part of a dog, the remaining tokens must complete the dog and thus are more constrained. Empirical evidence supports our motivation: at later generation stages, the next tokens can be well predicted by a multilayer perceptron, exhibit low variance, and follow closer-to-straight-line denoising paths from noise to tokens. Based on our finding, we introduce diffusion step annealing (DiSA), a training-free method that gradually uses fewer diffusion steps as more tokens are generated, *e.g.*, using 50 steps at the beginning and gradually decreasing to 5 steps at later stages. Because DiSA is derived from our finding specific to diffusion in AR models, it is complementary to existing acceleration methods designed for diffusion alone. DiSA can be implemented in only a few lines of code on existing models, and albeit simple, achieves $5-10\times$ faster inference for MAR and Harmon and $1.4-2.5\times$ for FlowAR and xAR, while maintaining the generation quality.

## 1 Introduction

Recent autoregressive (AR) models introduce diffusion sampling to generate continuous tokens, such as MAR (Li et al., 2024), FlowAR (Ren et al., 2024), xAR (Ren et al., 2025), and Harmon (Wu et al., 2025), which significantly improves generation quality. As illustrated in Figure 1(a-d), these models take generated tokens as input and adopt a diffusion process to sample the next tokens.

Although the diffusion process yields higher image quality for autoregressve models, it suffers from low inference efficiency because tens of denosing steps are needed to generate each token. For example, MAR (Li et al., 2024) denoises 100 times while xAR (Ren et al., 2025) does 50 times. Our preliminary experiments show that the many-step diffusion process accounts for about 50% inference latency in MAR and 90% in xAR. Naively reducing the number of diffusion steps can accelerate these models but will significantly degrade generation quality. With 10 diffusion steps, the Fréchet Inception Distance (FID) of xAR-L on ImageNet 256×256 increases shapely from 1.28 to 8.6, and MAR-L even fails to generate meaningful images. We thus aim to address the efficiency issue.

We are motivated by the finding that as more tokens are generated, token distributions become more constrained, and tokens become easier to sample. In other words, early generation stages are reliant on stronger distribution modeling and token sampling, while late stages are less so.

Three pieces of empirical evidence are provided to support our finding. First, we train a multilayer perceptron (MLP) or repurpose the original model head, based on the hidden representation of generated tokens, to predict the outcomes of the diffusion process. As shown in Figure 2, in early stages of generation, the prediction is inaccurate and lacks details. But as more tokens are generated, the prediction becomes increasingly accurate, indicating that the AR model now provides stronger conditions for the diffusion head. Second, variance in diffusion sampling gradually decreases during generation, indicating that the distribution of the next token becomes increasingly constrained. Third,

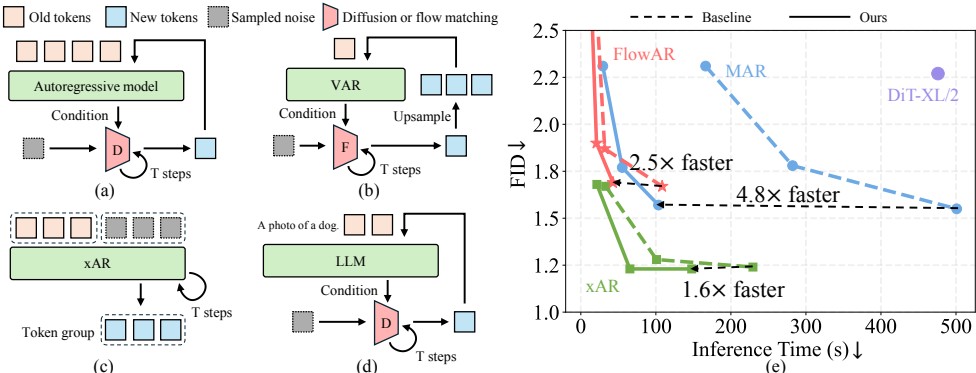

Figure 1: **Overview**. Architecture of four "AR + diffusion" models included in this study: (a) MAR (Li et al., 2024); (b) FlowAR (Ren et al., 2024); (c) xAR (Ren et al., 2025); (d) Harmon (Wu et al., 2025). (e) This paper improves the efficiency of these models by reducing diffusion steps gradually in the AR process without compromising generation quality.

based on the straightness metric (Liu et al., 2022b), we show that denoising paths from noise to tokens become closer to straight lines, suggesting that we could take larger step sizes.

The above finding dictates that fewer diffusion steps are needed in late generation stages than in early stages, forming the proposal of the diffusion step annealing (DiSA) method. Instead of using the same number of diffusion steps throughout the generation process, DiSA uses more diffusion steps (*e.g.*, 50) for early tokens and gradually fewer steps (*e.g.*, from 50 to 5) for later tokens.

DiSA is training-free and can be easily implemented on top of existing AR diffusion models that share similar token generation mechanisms, such as those in Figure 1(a-d). Moreover, because DiSA comes from our finding specific to diffusion in AR models, it can be effectively used together with existing acceleration methods specifically designed for diffusion. Experiments show that DiSA is very useful: it consistently improves the inference efficiency of MAR by $5 - 10\times$ and FlowAR and xAR by $1.4 - 2.5\times$ without sacrificing image generation quality.

In summary, this paper covers three main points. First, we reveal that the role of diffusion in AR models is different along the generation process. Second, based on this insight, we design a new sampling strategy, DiSA, for scheduling diffusion steps in AR image generation. Third, DiSA delivers significant inference acceleration while exhibiting competitive generation quality.

## 2 RELATED-WORK

**AR models meet diffusion**. A common practice for AR image generation is to quantize an image into discrete tokens (Esser et al., 2021; Razavi et al., 2019) and train AR models on the tokens (Sun et al., 2024; Chen et al., 2025; Wang et al., 2025). A main bottleneck is that discrete tokens introduce quantization errors, limiting the generation quality (Tschannen et al., 2023; Li et al., 2024; Han et al., 2024). To address this, MAR (Li et al., 2024) uses continuous tokens and adopts a diffusion model head to sample the next tokens in AR models. Other "AR + diffusion" design appears later (Ren et al., 2024; 2025; Wu et al., 2025). These methods have good generation quality but low efficiency.

**Acceleration techniques for diffusion models**. It is a well-established area in diffusion. Fast sampling processes have been proposed, such as DDIM (Song et al., 2020a), DPM-Solver (Lu et al., 2022b), and DPM-Solver++ (Lu et al., 2022a), to name a few. These methods are designed specifically for diffusion and can be used together with our approach. In comparison, less attention has been paid to accelerating diffusion in AR models. LazyMAR (Yan et al., 2025) introduces two caching techniques, while CSpD (Wang et al., 2024) applies speculative decoding for speeding up the inference of MAR. These works mainly focus on the AR part of MAR, without modifying the diffusion process, so are orthogonal to our approach. Besides, FAR (Hang et al., 2025) replaces the diffusion head of MAR with a short-cut model, achieving $2.3\times$ acceleration. Note that FAR is trained from scratch, while our method is training-free.

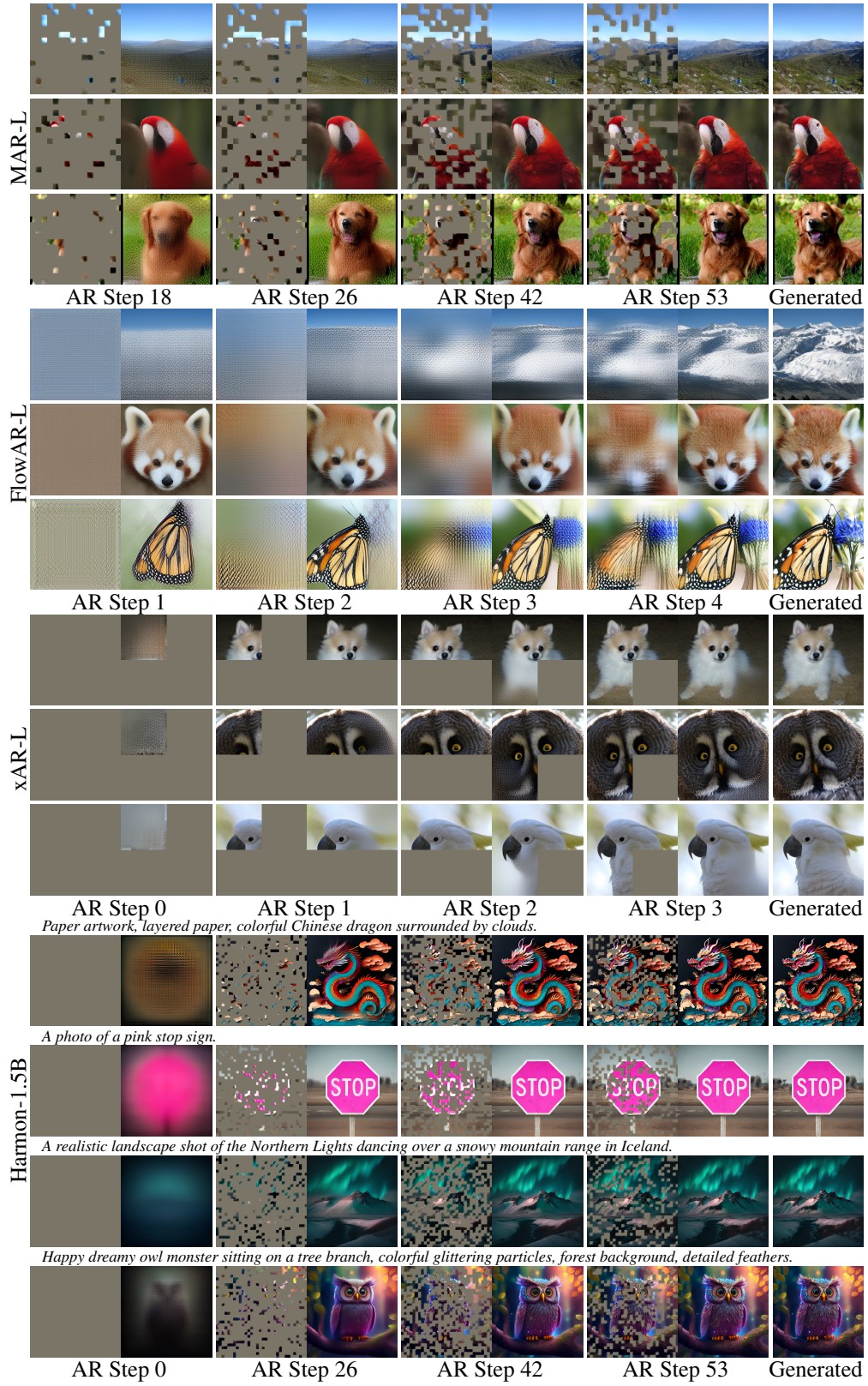

Figure 2: **Image prediction results** at different stages of generation. In each image pair, the left image shows the currently generated tokens, while the right shows the final image we predict based on the generated tokens. The prediction results are inaccurate and lack details in early stages but become increasingly accurate as more tokens are generated. This is consistent across the four models.

## 3 OBSERVATION ON AR + DIFFUSION MODELS

### 3.1 REVISITING EXISTING MODELS

With an image tokenizer, an image can be represented as a sequence of tokens $\langle \mathbf{x}^1, \mathbf{x}^2, \ldots, \mathbf{x}^n \rangle$. For example, we can use VAE (Kingma et al., 2013; Rombach et al., 2022b) to encode an image to 256 tokens. Image generation can be framed as sampling from the joint distribution of image tokens $p(\mathbf{x}^1, \mathbf{x}^2, \ldots, \mathbf{x}^n)$. Sampled tokens are decoded by the tokenizer back into images.

An AR model formulates the generation of an image as a next-token prediction task:

$$p(\mathbf{x}^1, \mathbf{x}^2, \ldots, \mathbf{x}^n) = \prod_{i=1}^{n} p(\mathbf{x}^i \mid \mathbf{x}^1, \ldots, \mathbf{x}^{i-1}) \text{ where } \mathbf{x}^i \sim p(\mathbf{x}^i \mid \mathbf{x}^1, \ldots, \mathbf{x}^{i-1}). \tag{1}$$

Note that new AR paradigms, such as next-scale prediction (Tian et al., 2024), generate a group of tokens in each AR step. For these models, $\mathbf{x}^i$ represents a group of tokens. We interchangeably use $\mathbf{x}^i$ to denote a single token or a group of tokens for simplicity.

Recent AR models adopt a diffusion process to sample $\mathbf{x}^i \sim p(\mathbf{x}^i \mid \mathbf{x}^1, \ldots, \mathbf{x}^{i-1})$.

**MAR** (Li et al., 2024) uses an encoder-decoder backbone $\mathbf{f}$, which takes generated tokens as input and predicts a condition vector $\mathbf{z}^i = \mathbf{f}(\mathbf{x}^1, \mathbf{x}^2, \ldots, \mathbf{x}^{i-1})$ for the next token. A diffusion model head $\epsilon_\theta$, conditional on $\mathbf{z}^i$, denoises noise to a token via reverse process. At training time, parameters in $\epsilon_\theta$ and $\mathbf{f}$ are updated based on the diffusion loss (Ho et al., 2020; Nichol & Dhariwal, 2021).

**FlowAR** (Liu et al., 2022b) uses VAR (Tian et al., 2024) as the backbone $\mathbf{f}$ and flow matching (Liu et al., 2022b; Ma et al., 2024) as the the model head $\mathbf{v}_\theta$. Similar to MAR, the backbone $\mathbf{f}$ takes tokens previous generated as input, and predicts a condition vector $\mathbf{z}^i$ for each next token. With a sampled noise token, the flow matching head predicts velocity for denoising the token. During training, the model is optimized with the flow matching loss (Lipman et al., 2022; Liu et al., 2022b).

**xAR** (Ren et al., 2025) takes both previously generated tokens and sampled noise as input. The model runs tens of times for denoising the noise into tokens and continues to sample the next tokens.

**Harmon** (Wu et al., 2025) is a unified model for both text-to-image (T2I) and image-to-text generation. This study focuses on its T2I ability. The backbone in Harmon takes the text prompt and generated tokens as input and produces a condition vector for the next token. A diffusion head, conditional on the vector, denoises sampled noise to the next token.

### 3.2 MORE TOKENS GENERATED, STRONGER CONSTRAINTS ON LATER TOKENS

The diffusion process in the four models samples the next token from the condition distribution $\mathbf{x}^i \sim p(\mathbf{x}^i \mid \mathbf{x}^1, \ldots, \mathbf{x}^{i-1})$. Our key motivation is that, as more tokens are generated, the condition becomes stronger, making the distribution more constrained and the next tokens easier to sample. We will show empirical evidence to support the motivation.

First, next tokens can be well predicted at later AR generation stages. We probe the condition from the generated tokens, *i.e.,* we use a model to predict the sampled $\mathbf{x}^i$ based on the hidden representation of the generated tokens $\{\mathbf{x}^1, \ldots, \mathbf{x}^{i-1}\}$. For MAR and Harmon, we train a MLP model to replace the original model head. The MLP predicts $\mathbf{x}^i$ directly based on the condition from the generated tokens $\mathbf{z}^i$. For FlowAR and xAR, we repurpose the original model head for flow matching. Specifically, we feed sampled noise with $t = 1$ into the model, obtain the estimated velocity $\mathbf{v}_\theta(\mathbf{x}_t^i \mid t = 1, \mathbf{z}^i)$, and predict the next token as $\mathbf{x}_0^i = \mathbf{x}_{t=1}^i - \mathbf{v}$. Since $\mathbf{x}_{t=1}^i$ is purely noisy, the model has to directly predict the $\mathbf{x}^i$ only based on the information in $\mathbf{z}^i$.

As shown in Figure 2, in the early stage of generation, the predicted tokens and the generated images are blurry and in low quality. But as more tokens have been generated, the MLP predictions become increasingly more accurate, suggesting that stronger conditions are provided by the generated tokens.

Second, next tokens have lower variance at later AR steps. We explore the variance in the distribution of the next tokens. Specifically, we use MAR to generate 10K images. When generating each $\mathbf{x}^i$, we sample 100 possible $\mathbf{x}^i$ and calculate the variance in sampling. The generated examples and the

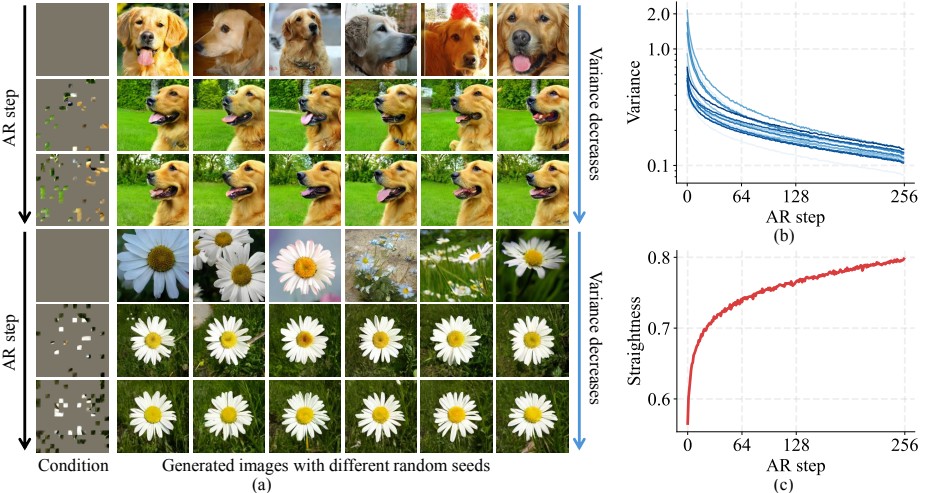

Figure 3: **Diffusion processes in later generation stages** show (a-b) lower variance and (c) closer-to-straight-line denoising paths. (a) Two examples. In each example, the AR step increases from top to bottom rows. 0%, 10%, 20% of tokens have been generated, respectively, as shown in the first column. We observe that the variance of sampled images drops from top to bottom rows. (b) Variance of diffusion-sampled tokens decreases along the AR steps. The y-axis uses a logarithmic scale and each line represents a different token dimension. (c) Straightness of denoising paths increases from early to late stages. All results are obtained from the MAR-B model.

average variance are shown in Figure 3(a-b). As seen, as more tokens are generated, the distribution of the next token becomes increasingly constrained.

Third, diffusion paths at later stages are closer to straight lines. Rectified Flow (Liu et al., 2022b) proposes that straight paths from noise to data distribution are preferred, because they can be simulated with coarse time discretization and hence need fewer steps at inference time. Inspired by this, we measure the straightness of denoising paths in each AR step. As shown in Figure 3(c), in the later stage of generation, the diffusion paths in MAR become closer to a straight line, indicating that we can use larger step sizes and fewer diffusion steps (Liu et al., 2022b). The results on FlowAR, xAR, and Harmon and details of implementation are shown in Section A.1.

### 3.3 Diffusion Step Annealing

Based on the observation, we propose a training-free sampling strategy, DiSA. In the early stage of generation, the distribution of the next tokens is diverse so we allow the diffusion process to run more times, *e.g.*, 50 steps. In the later stage, as the distribution of the next token is more constrained, we assign gradually fewer steps to diffusion, *e.g.*, 5 steps.

We introduce and compare three different time schedulers in DiSA: two-stage, linear, and cosine. Let $T(k)$ denote the number of diffusion steps when the AR step is $k$. $T_{early}$ and $T_{late}$ are two parameters to control the number of steps. In short, the two-stage method is just cutting the generation into the early and late stages. In the early stage, the diffusion process runs $T_{early}$ while in the late stage, runs $T_{late}$ times. The linear and cosine methods transition smoothly from $T_{early}$ to $T_{late}$ in the generation process. Specifically, they are defined as follows,

$$\text{Two-stage: } T(k) = \begin{cases} T_{early}, & k < K/2 \\ T_{late}, & \text{otherwise} \end{cases}, \tag{2}$$

$$\text{Linear: } T(k) = T_{early} + (T_{late} - T_{early}) \times k/K, \tag{3}$$

$$\text{Cosine: } T(k) = T_{late} + (T_{early} - T_{late}) \times \frac{1}{2} \left( \cos(\frac{k}{K}\pi) + 1 \right), \tag{4}$$

where $K$ is the total number of the AR steps and $T(k)$ is rounded to the nearest integer.

A preliminary experiment based on MAR is conducted to validate our method. We implement three time schedulers on MAR-B and MAR-L, modify values of $T_{early}$ and $T_{late}$, and evaluate the

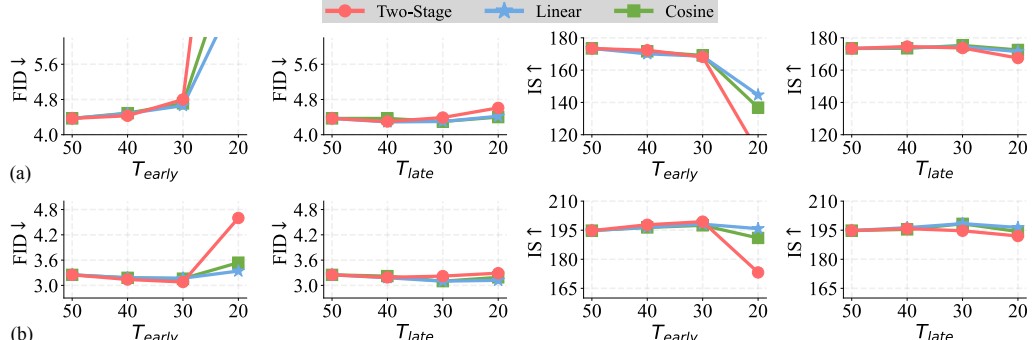

Figure 4: **Impact of different numbers of diffusion steps** in early generation stages $T_{early}$ and in late stages $T_{late}$ on (a) MAR-B; (b) MAR-L. In the first and third columns, we fix $T_{late} = 50$ and reduce $T_{early}$, which significantly degrades generation quality. But as shown in the second and fourth columns, if we fix $T_{early} = 50$ and decrease $T_{late}$, the degradation in generation quality is marginal.

model on ImageNet $256 \times 256$ generation. Fréchet Inception Distance (FID) (Heusel et al., 2017) and Inception scores (Salimans et al., 2016) on 50K sampled images are reported to measure the generation quality. The number of AR step is set to 64, and the default values of $T_{early}$ and $T_{late}$ are both 50. In Figure 4, reducing number of diffusion steps in early stages degrades the generation quality, but using fewer diffusion steps in later stages does not, which supports our motivation again. We use the linear scheduler in subsequent experiments, which has slightly better performance.

We find that reducing $T_{late}$ to less than 20 leads to poor generation results in MAR. The main reason is that the diffusion head has inaccurate prediction around $t = 999$. Thus, we let the diffusion start with $t = 950$, i.e., adding an initial time offset, following the practice in diffusion models (Song et al., 2020a; Liu et al., 2022a; Lin et al., 2024; von Platen et al., 2022). This allows us to further reduce the diffusion steps in MAR. For FlowAR and xAR, we do not observe this phenomenon and the sampling process starts with $t = 1.0$. We discuss this further in Section 4.2.

## 4 EXPERIMENTS

### 4.1 IMPLEMENTATION DETAILS, DATASETS, AND METRICS

Experiments mainly includes four pretrained models: MAR (Li et al., 2024), FlowAR (Ren et al., 2024), xAR (Ren et al., 2025), and Harmon (Wu et al., 2025). MAR, FlowAR, and xAR are evaluated on the ImageNet $256 \times 256$ generation task. We report FID (Heusel et al., 2017), IS (Salimans et al., 2016), Precision, and Recall, following common practice in image generation (Dhariwal & Nichol, 2021). We also measure the inference time of generating a batch of 256 images for these models. Harmon is evaluated on the T2I benchmark GenEval (Ghosh et al., 2023). Averaged accuracy and inference time are reported. All experiments are run on 4 NVIDIA A100 PCIe GPUs.

### 4.2 EVALUATION

**DiSA consistently improves the effiency of baseline AR+Diffusion models.** We apply DiSA to MAR, xAR, and FlowAR and compare the performance on the ImageNet $256 \times 256$ generation task in Table 1. Overall, DiSA consistently enhances the efficiency of the baseline models while maintaining competitive generation quality.

For MAR, the original best performance is achieved with 256 AR steps and 100 diffusion steps. After integrating DiSA to MAR, *e.g.,* $50 \rightarrow 5$, we report speed-up of $5.7\times$ on MAR-B, $5.1\times$ on MAR-L, and $4.8\times$ on MAR-H, respectively. The generation quality change is minor: DiSA results in the same FID on MAR-B and increases FID by 0.02 on MAR-H.

If we further reduce MAR to 32 AR steps and $25 \rightarrow 5$ diffusion steps, DiSA results in 9.3-11.3$\times$ speed-ups on MAR with slightly degraded generation quality. For example, DiSA achieves 11.3$\times$ faster inference on MAR-B while increasing FID by 0.04.

Table 1: **System-level method comparison** on ImageNet 256×256 Our method significantly improves the inference efficiency of MAR, FlowAR, and xAR, while maintaining their generation quality. Diffusion steps "$a \to b$" means starting with $a$ steps and transition to $b$ steps via Eq. (3). The average inference time per image and speed-ups of different methods are reported.

| | Model | #Params | AR steps | Diff steps | FID↓ | IS↑ | Pre.↑ | Rec.↑ | Time (s)↓ | Speed-Up↑ |
|---|---|---|---|---|---|---|---|---|---|---|
| Diff | LDM-4[†] (Rombach et al., 2022a) | 400M | - | - | 3.60 | 247.7 | 0.87 | 0.48 | - | - |
| | DiT-XL/2 (Peebles & Xie, 2023) | 675M | - | 250 | 2.27 | 278.2 | 0.83 | 0.57 | 1.859 | - |
| AR | GIVT (Tschannen et al., 2023) | 304M | 256 | - | 3.35 | - | 0.84 | 0.53 | - | - |
| | MAR-B (Li et al., 2024) | 208M | 256 | 100 | 2.31 | 281.7 | 0.82 | 0.57 | 0.650 | 1.0× |
| | | | 64 | 50 | 2.39 (+0.08) | 281.0 (-0.7) | 0.82 | 0.57 | 0.134 | 4.8× |
| | LazyMAR-B (Yan et al., 2025) | 208M | 64 | 100 | 2.45 (+0.14) | 281.3 (-0.4) | - | - | 0.061[*] | 10.6× |
| | | | 32 | 100 | 2.64 (+0.33) | 276.0 (-5.7) | - | - | 0.045[*] | 14.3× |
| | FAR-B (Hang et al., 2025) | 172M | 256 | 8 | 2.37 (+0.06) | 265.5 (-16.2) | - | - | - | 2.3× |
| | MAR-B + DiSA | 208M | 64 | 50→5 | 2.31 (+0.00) | 282.3 (+0.6) | 0.83 | 0.56 | 0.114 | 5.7× |
| | | | 32 | 25→5 | 2.35 (+0.04) | 282.9 (+1.2) | 0.83 | 0.56 | 0.057 | 11.3× |
| | MAR-L (Li et al., 2024) | 479M | 256 | 100 | 1.78 | 296.0 | 0.81 | 0.60 | 1.102 | 1.0× |
| | | | 64 | 50 | 1.86 (+0.08) | 294.0 (-2.0) | 0.80 | 0.61 | 0.250 | 4.4× |
| | LazyMAR-L (Yan et al., 2025) | 479M | 64 | 100 | 1.93 (+0.15) | 297.4 (+1.4) | - | - | 0.106[*] | 10.4× |
| | | | 32 | 100 | 2.11 (+0.33) | 284.4 (-11.6) | - | - | 0.080[*] | 13.8× |
| | FAR-L (Hang et al., 2025) | 406M | 256 | 8 | 1.99 (+0.21) | 293.0 (-3.0) | - | - | - | 1.4× |
| | MAR-L + CSpD (Wang et al., 2024) | - | - | - | 1.81 (+0.03) | 303.7 (+7.7) | - | - | - | 1.5× |
| | MAR-L + DiSA | 479M | 64 | 50→5 | 1.77 (-0.01) | 298.3 (+2.3) | 0.81 | 0.61 | 0.216 | 5.1× |
| | | | 32 | 25→5 | 1.88 (+0.10) | 295.1 (-0.9) | 0.81 | 0.61 | 0.108 | 10.2× |
| | MAR-H (Li et al., 2024) | 943M | 256 | 100 | 1.55 | 303.7 | 0.81 | 0.62 | 1.957 | 1.0× |
| | | | 64 | 50 | 1.65 (+0.10) | 299.8 (-3.9) | 0.80 | 0.62 | 0.462 | 4.2× |
| | LazyMAR-H (Yan et al., 2025) | 943M | 64 | 100 | 1.69 (+0.14) | 299.2 (-4.5) | - | - | 0.191[*] | 10.2× |
| | | | 32 | 100 | 1.94 (+0.39) | 284.1 (-19.6) | - | - | 0.145[*] | 13.5× |
| | MAR-H + CSpD (Wang et al., 2024) | - | - | - | 1.60 (+0.05) | 301.6 (-2.1) | - | - | - | 2.3× |
| | MAR-H + DiSA | 943M | 64 | 50→5 | 1.57 (+0.02) | 303.1 (-0.6) | 0.80 | 0.62 | 0.404 | 4.8× |
| | | | 32 | 25→5 | 1.72 (+0.17) | 303.4 (-0.3) | 0.80 | 0.61 | 0.209 | 9.3× |
| VAR | VAR-d30 (Tian et al., 2025) | 2.0B | 10 | - | 1.92 | 323.1 | 0.82 | 0.59 | 0.039[†] | - |
| | FlowAR-S (Ren et al., 2024) | 170M | 5 | 25 | 3.70 | 235.1 | 0.81 | 0.51 | 0.024 | 1.0× |
| | FlowAR-S + DiSA | 170M | 5 | 25→15 | 3.74 (+0.04) | 235.2 (+0.01) | 0.81 | 0.51 | 0.018 | 1.4× |
| | FlowAR-L (Ren et al., 2024) | 589M | 5 | 25 | 1.87 | 273.1 | 0.80 | 0.62 | 0.124 | 1.0× |
| | FlowAR-L + DiSA | 589M | 5 | 25→15 | 1.90 (+0.03) | 274.8 (+1.7) | 0.80 | 0.61 | 0.082 | 1.5× |
| | FlowAR-H (Ren et al., 2024) | 1.9B | 5 | 50 | 1.67 | 276.3 | 0.80 | 0.62 | 0.423[†] | 1.0× |
| | FlowAR-H + DiSA | 1.9B | 5 | 50→15 | 1.69 (+0.02) | 273.8 (-2.5) | 0.80 | 0.62 | 0.167[†] | 2.5× |
| xAR | xAR-B (Ren et al., 2025) | 172M | 4 | 50 | 1.67 | 265.2 | 0.80 | 0.62 | 0.130 | 1.0× |
| | xAR-B + DiSA | 172M | 4 | 50→15 | 1.68 (+0.01) | 265.5 (+0.3) | 0.79 | 0.62 | 0.084 | 1.6× |
| | xAR-L (Ren et al., 2025) | 608M | 4 | 50 | 1.28 | 292.5 | 0.82 | 0.62 | 0.394 | 1.0× |
| | xAR-L + DiSA | 608M | 4 | 50→15 | 1.23 (-0.05) | 287.3 (-5.2) | 0.79 | 0.66 | 0.255 | 1.5× |
| | xAR-H (Ren et al., 2025) | 1.1B | 4 | 50 | 1.24 | 301.6 | 0.83 | 0.64 | 0.896[†] | 1.0× |
| | xAR-H + DiSA | 1.1B | 4 | 50→15 | 1.23 (-0.01) | 300.5 (-1.1) | 0.79 | 0.66 | 0.577[†] | 1.6× |

[†] We test the latency of generating a batch of 128 images instead of 256 to reduce memory usage. [*] Estimated based on their paper.

Table 2: **Text-to-image generation** of Harmon on GenEval benchmark. The accuracy on each task and the average inference time per image are reported.

| AR steps | Diff steps | Single Obj. | Two Obj. | Counting | Colors | Position | Color Attri. | Overall | Time per image (s) |
|---|---|---|---|---|---|---|---|---|---|
| 32 | 50 | 0.99 | 0.86 | 0.64 | 0.87 | 0.43 | 0.49 | 0.71 | 12 |
| | 50→5 | 0.99 | 0.85 | 0.69 | 0.86 | 0.48 | 0.52 | 0.73 | 8 |
| 64 | 25 | 0.05 | 0.00 | 0.00 | 0.00 | 0.00 | 0.00 | 0.01 | 17 |
| | 25→5 | 0.99 | 0.89 | 0.74 | 0.86 | 0.46 | 0.54 | 0.75 | 14 |
| | 50 | 0.99 | 0.88 | 0.71 | 0.88 | 0.48 | 0.53 | 0.74 | 24 |
| | 50→5 | 0.99 | 0.89 | 0.68 | 0.87 | 0.41 | 0.55 | 0.73 | 17 |
| | 100 | 0.99 | 0.86 | 0.69 | 0.89 | 0.48 | 0.50 | 0.73 | 40 |
| | 100→5 | 0.99 | 0.90 | 0.68 | 0.86 | 0.49 | 0.51 | 0.74 | 24 |

Similarly, FlowAR-H with DiSA achieves a 2.5× speed-up while maintaining a competitive FID of 1.69 and IS of 273.8. In the case of xAR models, DiSA provides up to 1.6× speed-up with negligible impact on performance metrics. Interestingly, xAR-L shows 1.6× speed-up and even improved FID from 1.28 to 1.23 with DiSA. These results clearly indicate the usefulness of DiSA.

**Comparison with other acceleration methods on MAR.** DiSA is faster than CSpD (Wang et al., 2024) and FAR (Hang et al., 2025), and is competitive to LazyMAR (Yan et al., 2025). Note that LazyMAR works on caching techniques for MAR, without modifying the diffusion process, and is orthogonal to DiSA. It is interesting to combine LazyMAR and DiSA in future work.

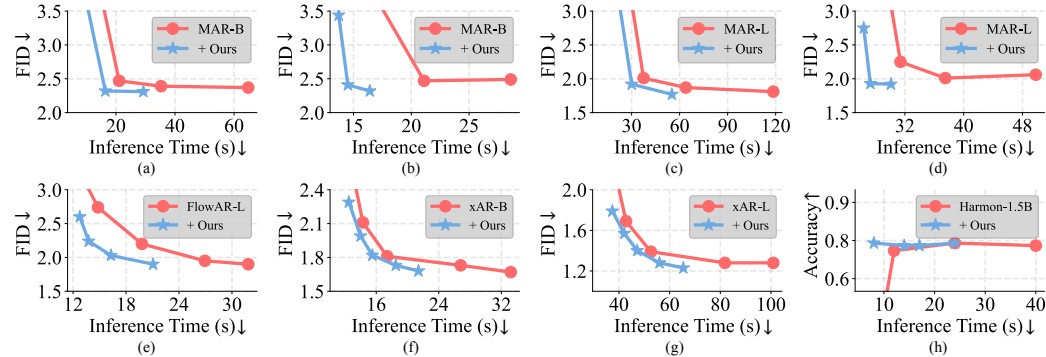

Figure 5: **Speed-quality trade-off** for (a) MAR-B with {16, 32, 64, 128} AR steps; (b) MAR-B with {25, 50, 100} diffusion steps; (c) MAR-L with {16, 32, 64, 128} AR steps; (d) MAR-L with {25, 50, 100} diffusion steps; (e) FlowAR-L with {8, 10, 15, 20, 25 } flow steps; (f) xAR-B and (g) xAR-L with {15, 20, 25, 40, 50} flow steps; and (h) Harmon-1.5B with different AR and diffusion steps.

**DiSA is also useful on T2I generation models.** As shown in Table 2, DiSA can also speed up Harmon on T2I generation tasks on GenEval. As seen, Harmon with DiSA uses 8 seconds per image, $5\times$ faster than the original implementation, while achieving a comparable performance.

**DiSA is complementary to existing diffusion acceleration methods.** We implement several existing diffusion acceleration techniques on MAR-B. **Time offset:** We start the diffusion process from $t = 950$ instead of $t = 999$. **Faster samplers:** We include DDIM (Song et al., 2020a), DPM-Solver (Lu et al., 2022b), and DPM-Solver++ (Lu et al., 2022a). Note that FlowAR uses the Euler sampler while xAR uses the Euler-Maruyama sampler (Maruyama, 1955; Higham, 2001), so we omit the detailed discussions of the two samplers here.

As shown in Table 3, existing techniques designed for diffusion can accelerate sampling in AR models. Time offset reduces the number of diffusion steps but suffers from a slight quality degradation. DDIM achieves a remarkable FID of 4.06 at 50 steps and 4.16 at 25 steps. DPM-Solver and DPM-Solver++ show comparable performance and reduce the number of diffusion steps to 25.

Table 3: Existing methods speed up MAR sampling and can be used together with DiSA for further speed-up. The number of AR steps is 64.

| Method | #Steps | FID↓ | IS↑ | Time (s)↓ |
|---|---|---|---|---|
| Original | 25 | 6.78 | 148.8 | 17.0 |
| | 50 | 4.30 | 174.5 | 21.9 |
| | 100 | 4.38 | 173.7 | 30.6 |
| Time offset | 25 | 4.61 | 171.0 | 16.8 |
| | 50 | 4.64 | 171.1 | 20.7 |
| + DiSA | 50→5 | 4.17 | 173.7 | 17.0 |
| DDIM | 25 | 4.16 | 178.2 | 17.7 |
| | 50 | 4.06 | 176.6 | 22.1 |
| + DiSA | 50→5 | 4.00 | 179.3 | 17.9 |
| DPM-Solver | 15 | 4.58 | 179.4 | 17.4 |
| | 25 | 4.35 | 176.1 | 20.6 |
| + DiSA | 25→10 | 4.37 | 177.1 | 17.9 |
| DPM-Solver++ | 15 | 4.57 | 179.5 | 18.5 |
| | 25 | 4.34 | 176.1 | 22.0 |
| + DiSA | 25→10 | 4.37 | 177.2 | 19.0 |

Our method is complementary to these diffusion acceleration approaches. If we combine time offset with DiSA, inference time can be reduced to 17.0 and FID is improved to 4.17. With a similar inference speed, time offset uses 25 steps and FID is 4.61. For the other three solvers, combining with DiSA also improves the inference speed while maintaining a comparable generation quality.

**Trade-off between efficiency and quality.** We show the trade-off of speed and generation quality in Figure 5. For MAR-B and MAR-L, we evaluate different AR and diffusion steps. FlowAR-L, xAR-B, and xAR-L are evaluated with different flow matching steps. Harmon-1.5B runs with different AR and diffusion steps on the GenEval benchmark. As seen, under different settings, DiSA can significantly improve the inference speed of these models, while maintaining the generation quality. We also present sample generation results in Figure 6. More results are provided in Section A.2.

## 4.3 DISCUSSION

**MAR vs MAE.** We show that an MLP can well predict the remaining tokens. This bridges the underlying mechanism between MAR and masked auto-encoder (MAE) (He et al., 2022). The former uses a generative method to unmask an image, while the latter uses a deterministic way to do so. This is also consistent with recent findings where MAR encodes semantic information for an image (Wu et al., 2025). See Section A.3 for further discussion.

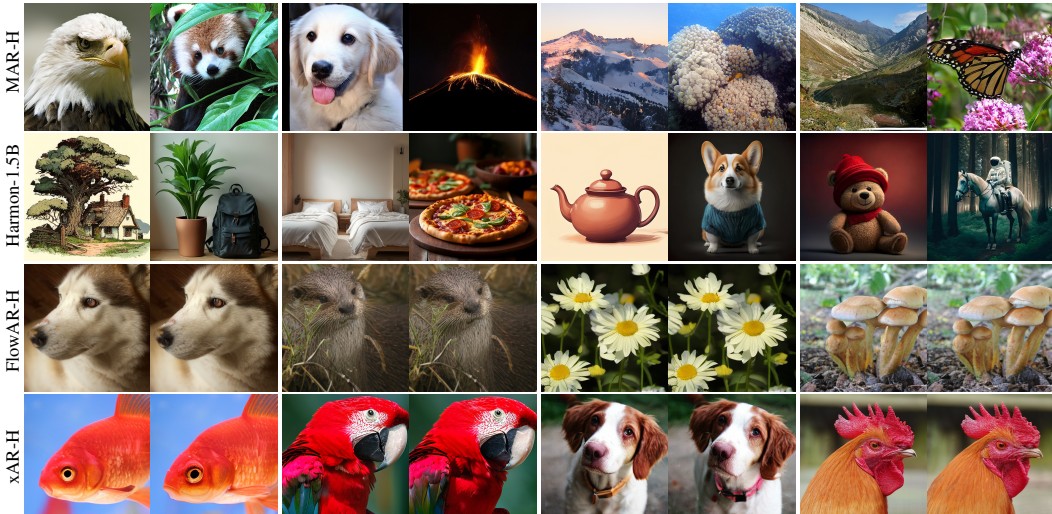

Figure 6: **Sample image generation results.** For MAR-H and Harmon-1.5B, we present the samples generated using DiSA. For FlowAR and xAR, each image pair is generated with the same random seed, where the first is generated without DiSA while the other is with DiSA. We find that DiSA helps generate similar quality images while speeding up image generation by $2.5\times$ and $1.6\times$ respectively.

**Difficulty level of token distribution modeling.** Condition vectors in later generation stages offer more information, making token distributions easier to model. This may also hold in other AR models. For example, recent works use Gaussian Mixture Model to model token distribution (Tschannen et al., 2023; Zhao et al., 2025). It is possible that the early stage needs more Gaussian components while later stages require fewer. We leave this as future work.

**Heuristics for Adjusting Diffusion Step Shedule** Automatically adjusting the denoising schedule in DiSA is a challenging but promising research direction. As a preliminary study, we explore three heuristics to guide this process, based on our empirical experiments: the straightness of denoising paths, the variance of diffusion-sampled tokens, and the uncertainty in predicting mask tokens. These heuristics can be applied in two ways. (1) Offline. a fixed schedule is designed after analyzing the heuristic values from 50,000 generated images. (2) Online. the heuristics dynamically adjust the number of diffusion steps for subsequent tokens based on the current generation process. Our findings indicate that these heuristics lead to comparable performance, with the uncertainty heuristic achieving a $6.0\times$ speed-up for MAR-B while maintaining similar quality.

**Strong diffusion conditions in computer vision.** It is intuitive to understand that the condition vector which summarizes more previously generated tokens is more informative. Therefore, fewer diffusion steps would still sample a good token. This is consistent with some existing works in image contour detection and depth estimation using diffusion models: because of the strong image condition, a few and even one diffusion step would yield competitive results (Liu et al., 2025; Song et al., 2025; Zhou et al., 2024). In T2I generation, a text prompt seems a weak condition. Our prediction results on Harmon in Figure 2 show that, a text prompt helps to determine the basic the structure of the image, for example, the shape and color of the stop sign, leaving details for generation.

## 5 CONCLUSION

To sum up, this paper studies how to effectively reduce the number of diffusion steps in AR models. We find that as more tokens are generated, the reliance on many diffusion steps is alleviated. Based on this, we propose DiSA, a training-free strategy that gradually decreases the number of diffusion steps during the AR generation process. This approach is easy to implement and significantly improves inference speed while maintaining competitive image quality. Our study provides interesting insights into the diffusion process in AR image generation, and our future work will investigate how our insights can be generalized and applied to other AR models.

ETHICS STATEMENT

Our research focuses on improving the efficiency of image generation models while maintaining image quality. While we do not propose a new generation algorithm, we acknowledge that image generation models, in general, may pose potential societal risks, including the creation of misleading or synthetic content. Faster generation techniques may inadvertently lower the barrier for malicious use. Possible mitigation strategies include integrating safeguards such as deepfake detection systems, introducing watermarking techniques, and developing responsible use guidelines.

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
