# A APPENDIX

## A.1 IMPLEMENTATION DETAILS

We download the pretrained model checkpoints that the original authors provided.

- MAR (Li et al., 2024). https://github.com/LTH14/mar
- FlowAR (Ren et al., 2024). https://github.com/OliverRensu/FlowAR
- xAR (Ren et al., 2025). https://github.com/OliverRensu/xAR
- Harmon (Wu et al., 2025). https://github.com/wusize/Harmon

**Predict the Next Tokens.** To predict the next tokens, we train a MLP on MAR and Harmon. The MLP has a similar depth and width as the original model head. For example, on MAR-H, we use a MLP with 12 blocks and a width of 1536 channels. For MAR, we freeze the pretrained model and train the MLP on the ImageNet 256×256 dataset.

In training, each batch of images is converted to tokens, and a random ratio of tokens are masked. The unmasked tokens are fed into the encoder-decoder part of MAR. The decoder outputs a condition vector for each mask token. The MLP takes the condition vector as input and predict the mask token with MSE loss.

For Harmon, we use a distillation strategy to train the MLP. Specifically, we use GenEval (Ghosh et al., 2023) to create 713 prompts, and use the Harmon model to generates images under each prompt. We train an MLP in the process to predict the generated tokens.

At inference time, when MAR or Harmon generates a few tokens, we hack in the model to get the condition vectors for the remaining tokens, and use our trained MLP to predict the remaining tokens. Then, we feed generated tokens and predicted tokens to the tokenizer to get the images.

For FlowAR and xAR, we repurpose the original model head. As described in the main context, we feed sampled noise with $t = 1$ into the model, obtain the estimated velocity $\mathbf{v}_\theta(\mathbf{x}_t^i \mid t = 1, \mathbf{z}^i)$, and predict the next token as $\mathbf{x}_0^i = \mathbf{x}_{t=1}^i - \mathbf{v}$. Since $\mathbf{x}_{t=1}^i$ is purely noisy, the model has to directly predict the $\mathbf{x}^i$ based on $\mathbf{z}^i$. This method is training-free. For MAR and Harmon, the model predictions on $t = 999$ are unreliable, thus, we train MLPs instead.

At inference time, when FlowAR or xAR generates a few tokens, we change the original model head to predict the remaining tokens in one step, and get the predicted images.

**Variance and Straightness of Diffusion Processes in FlowAR, xAR, and Harmon.** we measure the straightness of a denoising path $\{\mathbf{x}_t\}_{t=0}^1$ under condition $\mathbf{z}$.

$$S(\{\mathbf{x}_t\}_{t=0}^1, \mathbf{z}) = \mathbb{E}_{t \sim [0,1]} \left[ \|(\mathbf{x}_1 - \mathbf{x}_0) - \mathbf{v}_\theta(\mathbf{x}_t \mid t, \mathbf{z})\|^2 \right]. \tag{5}$$

MAR and Harmon use diffusion process and are not trained on the rectified flow loss function. Thus, we calculate the cosine similarity between the score (the gradient of the data distribution density) (Song & Ermon, 2019; Song et al., 2020b; Dhariwal & Nichol, 2021) and the straight direction from the noisy token to the clean token.

$$S(\{\mathbf{x}_t\}_{t=0}^{999}, \mathbf{z}) = \mathbb{E}_t \left[ \cos \left( \mathbf{x}_0 - \mathbf{x}_t, \nabla_{\mathbf{x}_t} \log p_\theta(\mathbf{x}_t \mid t, \mathbf{z}) \right) \right], \tag{6}$$

where $\nabla_{\mathbf{x}_t} \log p_\theta(\mathbf{x}_t \mid t, \mathbf{z}) = -\frac{1}{\sqrt{1 - \bar{\alpha}_t}} \boldsymbol{\epsilon}_\theta(\mathbf{x}_t \mid t, \mathbf{z})$.

Variance and straightness of FlowAR, xAR and Harmon are shown in Figure 7. Note that FlowAR and xAR are trained with the Flow Matching loss, which explicitly requires the flow paths to be straight. Thus, the straightness of FlowAR and xAR is much higher than that of the other two models. But the two models show a similar trend: not so straight at the beginning but increase significantly at later stages.

An interesting finding is that FlowAR is not exactly following our hypothesis. The straightness peaks in the middle, and decreases slightly at the end of generation. This may inspire a different strategy on FlowAR, that is, use fewer steps in the middle. It is also possible to consider a straightness-aware adaptive method, which considers the straightness of different models to schedule the number of diffusion steps. We leave these as future work.

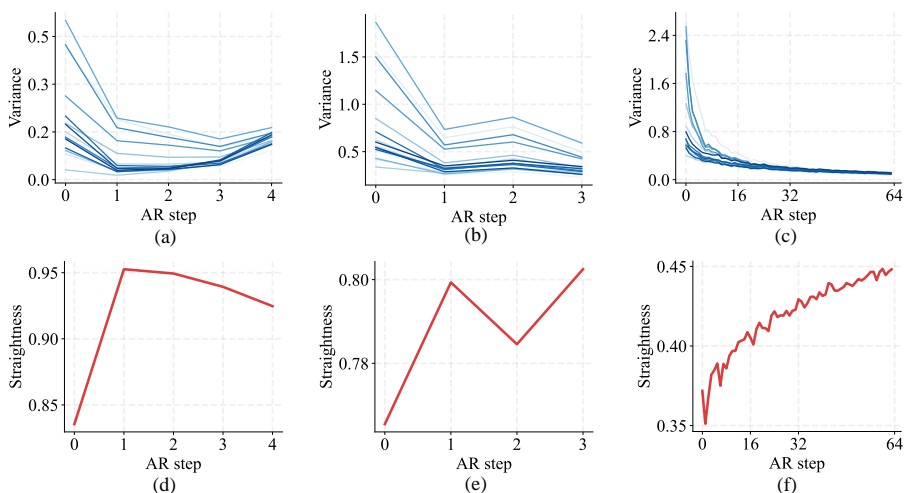

Figure 7: Variance of diffusion-sampled tokens decreases along the autoregressive steps in (a) FlowAR, (b) xAR, and (c) Harmon. Besides, straightness of denoising paths increases from early to late stages in (d) FlowAR, (e) xAR, and (f) Harmon.

Table 4: **Speed-Quality Trade-off** in MAR-B. Results are partly visualized in Figure 5.

| AR steps | 8 | 16 | 32 | 32 | 32 | 64 | 128 |
|---|---|---|---|---|---|---|---|
| Diff steps | 50 | 50 | 25 | 50 | 100 | 50 | 50 |
| FID↓ | 12.41 | 3.92 | 3.65 | 2.47 | 2.49 | 2.39 | 2.37 |
| IS↑ | 184.6 | 250.3 | 240.7 | 272.4 | 273.8 | 280.4 | 279.0 |
| Time(s)↓ | 10.5 | 14.0 | 17.1 | 21.1 | 28.6 | 35.3 | 64.8 |

**Implementation of DiSA.** DiSA can be implemented within a few lines of code. We show a pseudo-code for showing how DiSA works on top of existing models. Before running a diffusion / flow matching process to sample the next tokens, we calculate $T_k$ first and get the time schedule with $T_k$ steps.

---
**Algorithm 1** Conditional Token Generation via Diffusion with Adaptive Steps
---
1: **function** GENERATETOKEN($z$: condition; $k$: current AR step; $K$: the total number of AR steps)
2:     Get $T_k$ via Eq. (3)
3:     $\{t_i\}_{i=1}^{T_k} \leftarrow$ GETTIMESCHEDULE($T_k$)
4:     $x \leftarrow$ SAMPLEGAUSSIANNOISE
5:     **for** $t$ in $\{t_i\}$ **do**
6:         $x \leftarrow$ DENOISESTEP($x, t, z$)
7:     **end for**
8:     **return** $x$
9: **end function**

---

## A.2 DETAILED RESULTS AND MORE EXAMPLES ON SPEED-QUALITY TRADE-OFF

### A.2.1 SPEED-QUALITY TRADE-OFF

We report detailed results for Figure 5 in Tables 4-13.

### A.2.2 CHOICE ACCORDING TO AUTOREGRESSIVE STEPS.

As seen in Table 14, using fewer autoregressive steps in MAR significantly degrades generation quality. In contrast, DiSA can often maintain or even improve the quality. Besides, DiSA is not fragile but works well across a wide range of $T_{late}$ values.

Table 5: **Speed-Quality Trade-off** in MAR-B + DiSA. Results are partly visualized in Figure 5.

| AR steps | 8 | 16 | 32 | 32 | 32 | 64 |
|---|---|---|---|---|---|---|
| Diff steps | 50→5 | 50→5 | 10→5 | 20→5 | 50→5 | 50→5 |
| FID↓ | 10.97 | 3.66 | 3.44 | 2.41 | 2.32 | 2.31 |
| IS↑ | 196.3 | 255.1 | 279.1 | 283.4 | 279.3 | 282.3 |
| Time(s)↓ | 6.55 | 9.92 | 13.72 | 14.51 | 16.41 | 29.3 |

Table 6: **Speed-Quality Trade-off** in MAR-L. Results are partly visualized in Figure 5.

| AR steps | 8 | 16 | 32 | 32 | 32 | 64 | 128 |
|---|---|---|---|---|---|---|---|
| Diff steps | 50 | 50 | 25 | 50 | 100 | 50 | 50 |
| FID↓ | 14.70 | 4.03 | 2.25 | 2.01 | 2.06 | 1.87 | 1.81 |
| IS↑ | 173.8 | 253.3 | 278.3 | 285.4 | 283.1 | 294.9 | 298.7 |
| Time(s)↓ | 17.4 | 24.0 | 31.4 | 37.5 | 49.8 | 63.9 | 118.5 |

Table 7: **Speed-Quality Trade-off** in MAR-L + DiSA. Results are partly visualized in Figure 5.

| AR steps | 8 | 16 | 32 | 32 | 32 | 64 |
|---|---|---|---|---|---|---|
| Diff steps | 50→5 | 50→5 | 10→5 | 20→5 | 50→5 | 50→5 |
| FID↓ | 13.65 | 3.86 | 2.75 | 1.93 | 1.92 | 1.77 |
| IS↑ | 181.0 | 251.9 | 294.8 | 295.8 | 285.4 | 298.3 |
| Time(s)↓ | 11.0 | 17.5 | 26.4 | 27.3 | 30.1 | 55.2 |

Table 8: **Speed-Quality Trade-off** in FlowAR-L. Results are partly visualized in Figure 5.

| AR steps | 5 | 5 | 5 | 5 | 5 |
|---|---|---|---|---|---|
| Diff steps | 8 | 10 | 15 | 20 | 25 |
| FID↓ | 3.37 | 2.74 | 2.20 | 1.95 | 1.90 |
| IS↑ | 294.9 | 293.9 | 287.2 | 275.9 | 281.4 |
| Time(s)↓ | 12.4 | 14.8 | 19.8 | 26.9 | 31.8 |

Table 9: **Speed-Quality Trade-off** in FlowAR-L + DiSA. Results are partly visualized in Figure 5.

| AR steps | 5 | 5 | 5 | 5 |
|---|---|---|---|---|
| Diff steps | 15→8 | 20→8 | 25→10 | 25→15 |
| FID↓ | 2.60 | 2.24 | 2.03 | 1.90 |
| IS↑ | 280.6 | 274.2 | 274.3 | 274.8 |
| Time(s)↓ | 12.8 | 13.8 | 16.3 | 21.1 |

Table 10: **Speed-Quality Trade-off** in xAR-B. Results are partly visualized in Figure 5.

| AR steps | 4 | 4 | 4 | 4 | 4 |
|---|---|---|---|---|---|
| Diff steps | 15 | 20 | 25 | 40 | 50 |
| FID↓ | 3.16 | 2.11 | 1.81 | 1.73 | 1.67 |
| IS↑ | 247.6 | 258.5 | 264.7 | 266.3 | 265.2 |
| Time(s)↓ | 11.2 | 14.3 | 17.4 | 26.8 | 33.2 |

Table 11: **Speed-Quality Trade-off** in xAR-B + DiSA. Results are partly visualized in Figure 5.

| AR steps | 4 | 4 | 4 | 4 | 4 |
|---|---|---|---|---|---|
| Diff steps | 25→10 | 25→15 | 30→15 | 40→15 | 50→15 |
| FID↓ | 2.29 | 1.99 | 1.82 | 1.73 | 1.68 |
| IS↑ | 253.6 | 259.3 | 263.1 | 262.4 | 265.5 |
| Time(s)↓ | 12.5 | 14.0 | 15.5 | 18.5 | 21.4 |

Table 12: **Speed-Quality Trade-off** in xAR-L. Results are partly visualized in Figure 5.

| AR steps | 4 | 4 | 4 | 4 | 4 |
|---|---|---|---|---|---|
| Diff steps | 15 | 20 | 25 | 40 | 50 |
| FID↓ | 2.79 | 1.69 | 1.39 | 1.28 | 1.28 |
| IS↑ | 260.3 | 278.7 | 286.0 | 292.2 | 292.2 |
| Time(s)↓ | 33.0 | 42.8 | 52.6 | 81.8 | 100.9 |

Table 13: **Speed-Quality Trade-off** in xAR-L + DiSA. Results are partly visualized in Figure 5.

| AR steps | 4 | 4 | 4 | 4 | 4 |
|---|---|---|---|---|---|
| Diff steps | 25→10 | 25→15 | 30→15 | 40→15 | 50→15 |
| FID↓ | 1.79 | 1.57 | 1.40 | 1.28 | 1.23 |
| IS↑ | 275.8 | 280.0 | 284.2 | 292.4 | 287.3 |
| Time(s)↓ | 37.4 | 42.1 | 47.2 | 56.0 | 65.4 |

Table 14: **Performance of MAR-B and MAR-L** with DiSA under different AR steps and diffusion step schedules.

| Model | AR steps | Diff steps | FID↓ | IS↑ | Time (s)↓ | Speed-Up↑ |
|---|---|---|---|---|---|---|
| MAR-B | 256 | 100 | 2.31 | 281.7 | 0.650 | 1.0× |
| | 64 | 50 | 2.39 (+0.08) | 281.0 (-0.7) | 0.134 | 4.8× |
| +DiSA | 64 | 50→25 | 2.31 (+0.00) | 278.9 (-2.8) | 0.126 | 5.2× |
| | 64 | 50→15 | 2.26 (-0.05) | 281.0 (-0.7) | 0.120 | 5.4× |
| | 64 | 50→10 | 2.29 (-0.02) | 279.3 (-2.4) | 0.119 | 5.5× |
| | 64 | 50→5 | 2.31 (+0.00) | 282.3 (+0.6) | 0.114 | 5.7× |
| | 32 | 50→25 | 2.56 (+0.25) | 272.0 (-9.7) | 0.071 | 9.2× |
| | 32 | 50→15 | 2.47 (+0.16) | 272.0 (-9.7) | 0.067 | 9.7× |
| | 32 | 50→10 | 2.42 (+0.11) | 273.9 (-7.8) | 0.065 | 10.0× |
| | 32 | 50→5 | 2.32 (+0.01) | 279.3 (-2.4) | 0.063 | 10.4× |
| | 16 | 50→25 | 4.33 (+2.02) | 246.4 (-35.3) | 0.045 | 14.4× |
| | 16 | 50→15 | 4.15 (+1.84) | 247.8 (-33.9) | 0.042 | 15.6× |
| | 16 | 50→10 | 4.01 (+1.70) | 249.5 (-32.2) | 0.040 | 16.3× |
| | 16 | 50→5 | 3.65 (+1.34) | 255.2 (-26.5) | 0.038 | 16.9× |
| | 8 | 50→25 | 13.59 (+11.28) | 179.6 (-102.1) | 0.031 | 20.7× |
| | 8 | 50→15 | 13.15 (+10.84) | 182.9 (-98.8) | 0.029 | 22.5× |
| | 8 | 50→10 | 12.62 (+10.31) | 185.8 (-95.9) | 0.027 | 23.8× |
| | 8 | 50→5 | 10.97 (+8.66) | 196.2 (-85.5) | 0.026 | 25.5× |
| MAR-L | 256 | 100 | 1.78 | 296.0 | 1.102 | 1.0× |
| | 64 | 50 | 1.86 (+0.08) | 294.0 (-2.0) | 0.250 | 4.4× |
| +DiSA | 64 | 50→25 | 1.83 (+0.05) | 290.1 (-5.9) | 0.232 | 4.8× |
| | 64 | 50→15 | 1.78 (+0.00) | 293.3 (-2.7) | 0.224 | 4.9× |
| | 64 | 50→10 | 1.80 (+0.02) | 292.3 (-3.7) | 0.225 | 4.9× |
| | 64 | 50→5 | 1.77 (-0.01) | 298.3 (+2.3) | 0.216 | 5.1× |
| | 32 | 50→25 | 2.21 (+0.43) | 278.6 (-17.4) | 0.129 | 8.5× |
| | 32 | 50→15 | 2.14 (+0.36) | 281.6 (-14.4) | 0.123 | 9.0× |
| | 32 | 50→10 | 2.06 (+0.28) | 281.2 (-14.8) | 0.120 | 9.2× |
| | 32 | 50→5 | 1.92 (+0.14) | 285.5 (-10.5) | 0.117 | 9.4× |
| | 16 | 50→25 | 4.72 (+2.94) | 243.7 (-52.3) | 0.079 | 13.9× |
| | 16 | 50→15 | 4.52 (+2.74) | 245.4 (-50.6) | 0.074 | 14.9× |
| | 16 | 50→10 | 4.31 (+2.53) | 247.7 (-48.3) | 0.071 | 15.5× |
| | 16 | 50→5 | 3.86 (+2.08) | 251.8 (-44.2) | 0.069 | 16.0× |
| | 8 | 50→25 | 16.81 (+15.03) | 164.0 (-132.0) | 0.054 | 20.6× |
| | 8 | 50→15 | 16.34 (+14.56) | 164.5 (-131.5) | 0.050 | 22.3× |
| | 8 | 50→10 | 15.51 (+13.73) | 169.1 (-126.9) | 0.046 | 23.7× |
| | 8 | 50→5 | 13.64 (+11.86) | 181.0 (-115.0) | 0.044 | 25.0× |

Table 15: **Performance** of MAR with DiSA under raster and reverse raster orders.

| Model | AR steps | Diff steps | FID↓ | IS↑ |
|---|---|---|---|---|
| *Raster order* | | | | |
| MAR-B | 64 | 50 | 7.18 | 246.8 |
| +DiSA | 64 | 50→25 | 7.72 (+0.54) | 241.1 (-5.7) |
| | 64 | 50→15 | 7.57 (+0.39) | 241.0 (-5.8) |
| MAR-L | 64 | 50 | 8.13 | 236.4 |
| +DiSA | 64 | 50→25 | 9.81 (+1.68) | 221.3 (-15.1) |
| | 64 | 50→15 | 9.47 (+1.34) | 222.7 (-13.7) |
| *Reverse raster order* | | | | |
| MAR-B | 64 | 50 | 7.02 | 248.8 |
| +DiSA | 64 | 50→25 | 7.80 (+0.78) | 238.6 (-10.2) |
| | 64 | 50→15 | 7.53 (+0.51) | 241.6 (-7.2) |
| MAR-L | 64 | 50 | 10.82 | 202.4 |
| +DiSA | 64 | 50→25 | 12.87 (+2.05) | 188.1 (-14.3) |
| | 64 | 50→15 | 12.34 (+1.52) | 192.6 (-9.8) |

Table 16: **Performance** of MAR with DiSA under bfloat16.

| Model | AR steps | Diff steps | FID↓ | IS↑ |
|---|---|---|---|---|
| MAR-B | 64 | 50 | 2.39 | 281.0 |
| MAR-B (BF16) | 64 | 50 | 2.49 (+0.10) | 282.7 (+1.7) |
| +DiSA (BF16) | 64 | 50→25 | 2.31 (-0.08) | 278.3 (-2.7) |
| | 64 | 50→15 | 2.30 (-0.09) | 278.0 (-3.0) |
| | 64 | 50→10 | 2.30 (-0.09) | 279.6 (-1.4) |
| | 64 | 50→5 | 2.34 (-0.05) | 279.7 (-1.3) |
| MAR-L | 64 | 50 | 1.86 | 294.0 |
| MAR-L (BF16) | 64 | 50 | 1.87 (+0.01) | 292.7 (-1.3) |
| +DiSA (BF16) | 64 | 50→25 | 1.81 (-0.05) | 293.2 (-0.8) |
| | 64 | 50→15 | 1.81 (-0.05) | 289.5 (-4.5) |
| | 64 | 50→10 | 1.76 (-0.10) | 293.9 (-0.1) |
| | 64 | 50→5 | 1.77 (-0.09) | 295.1 (+1.1) |

### A.2.3 CHOICE ACCORDING TO AR GENERATION ORDER

In our experiments, MAR generates tokens in random orders. We mannually set MAR to generate tokens in raster order (from left to right and from up to bottom) or reverse raster order. As shown in Table 15, DiSA leads a slight performance drop with MAR. This may result from training-testing mismatch, since the pretrained MAR model is not optimized for raster order. On models pretrained with raster order like xAR, DiSA performs well.

### A.2.4 PERFORMANCE UNDER BFLOAT16

Many large models are trained and evaluated using bfloat16 (BF16) precision. We tested DiSA in this setting and found that MAR's performance drops under bfloat16, while DiSA still improves the generation quality. The results are shown in Table 16.

### A.2.5 HEURISTICS FOR ADJUSTING DIFFUSION STEP SHEDULE

Automatically adjusting the denoising schedule may be challenging and is an interesting research direction. Based on the three empirical experiments in the main content, we explore three heuristics to guide the scheduler.

1. Straightness of denoising paths.

2. Variance of diffusion-sampled tokens.

3. Uncertainty in predicting mask tokens.

There are two ways to use these heuristics:

1. Offline: We let the model generate 50K images, measure the heuristic values, and then design a fixed schedule based on that.

Table 17: **Heuristics** for adjusting the denoising step schedule.

| Model | AR steps | Diff steps | FID↓ | IS↑ | Time (s)↓ | Speed-Up↑ |
|-------|----------|-----------|------|-----|-----------|-----------|
| MAR-B | 256 | 100 | 2.31 | 281.7 | 0.650 | 1.0× |
| | 64 | 50 | 2.39 (+0.08) | 281.0 (-0.7) | 0.134 | 4.8× |
| +DiSA | 64 | 50→5 | 2.31 (+0.00) | 282.3 (+0.6) | 0.114 | 5.7× |
| +DiSA (Offline) | 64 | Straightness heuristics | 2.30 (-0.01) | 282.2 (+0.5) | 0.120 | 5.4× |
| | 64 | Variance heuristics | 2.31 (+0.00) | 283.0 (+1.3) | 0.108 | 6.0× |
| | 64 | Uncertainty heuristics | 2.30 (-0.01) | 283.9 (+2.2) | 0.109 | 6.0× |
| +DiSA (Online) | 64 | Straightness heuristics | 2.29 (-0.02) | 279.0 (-2.7) | 0.129 | 5.0× |
| | 64 | Variance heuristics | 2.40 (+0.09) | 281.2 (-0.5) | 0.138 | 4.7× |
| | 64 | Uncertainty heuristics | 2.30 (-0.01) | 282.2 (+0.5) | 0.109 | 6.0× |
| MAR-L | 256 | 100 | 1.78 | 296.0 | 1.102 | 1.0× |
| | 64 | 50 | 1.86 (+0.08) | 294.0 (-2.0) | 0.250 | 4.4× |
| +DiSA | 64 | 50→5 | 1.77 (-0.01) | 298.3 (+2.3) | 0.216 | 5.1× |
| +DiSA (Offline) | 64 | Straightness heuristics | 1.80 (+0.02) | 292.2 (-3.8) | 0.231 | 4.8× |
| | 64 | Variance heuristics | 1.81 (+0.03) | 292.8 (-3.2) | 0.222 | 5.0× |
| | 64 | Uncertainty heuristics | 1.78 (+0.00) | 295.3 (-0.7) | 0.198 | 5.6× |
| +DiSA (Online) | 64 | Straightness heuristics | 1.82 (+0.04) | 291.4 (-4.6) | 0.245 | 4.5× |
| | 64 | Variance heuristics | 1.81 (+0.03) | 291.6 (-4.4) | 0.261 | 4.2× |
| | 64 | Uncertainty heuristics | 1.78 (+0.00) | 295.2 (-0.8) | 0.198 | 5.6× |
| FlowAR-H | 5 | 50 | 1.67 | 276.3 | 0.423 | 1.0× |
| +DiSA | 5 | 50→15 | 1.69 (+0.02) | 273.8 (-2.5) | 0.167 | 2.5× |
| +DiSA (Offline) | 5 | Straightness heuristics | 1.70 (+0.03) | 275.3 (-1.0) | 0.189 | 2.2× |
| | 5 | Variance heuristics | 1.71 (+0.04) | 277.1 (+0.8) | 0.196 | 2.2× |
| | 5 | Uncertainty heuristics | 1.87 (+0.20) | 281.8 (+5.5) | 0.159 | 2.7× |
| +DiSA (Online) | 5 | Straightness heuristics | 1.71 (+0.04) | 276.0 (-0.3) | 0.197 | 2.1× |
| | 5 | Variance heuristics | 1.72 (+0.05) | 274.5 (-1.8) | 0.404 | 1.0× |
| | 5 | Uncertainty heuristics | 1.82 (+0.15) | 283.0 (+6.7) | 0.181 | 2.3× |
| xAR-H | 4 | 50 | 1.24 | 301.6 | 0.896 | 1.0× |
| +DiSA | 4 | 50→15 | 1.23 (-0.01) | 300.5 (-1.1) | 0.577 | 1.6× |
| +DiSA (Offline) | 4 | Straightness heuristics | 1.26 (+0.02) | 298.5 (-3.1) | 0.478 | 1.9× |
| | 4 | Variance heuristics | 1.27 (+0.03) | 299.2 (-2.4) | 0.495 | 1.8× |
| | 4 | Uncertainty heuristics | 1.24 (+0.00) | 298.8 (-2.8) | 0.521 | 1.7× |
| +DiSA (Online) | 4 | Straightness heuristics | 1.25 (+0.01) | 299.9 (-1.7) | 0.611 | 1.5× |
| | 4 | Variance heuristics | - | - | - | Too slow |
| | 4 | Uncertainty heuristics | 1.27 (+0.03) | 299.9 (-1.7) | 0.543 | 1.7× |

2. Online: During generation, the heuristics evaluate the generation process of the current tokens. For instance, if the variance of current tokens keeps decreasing, the scheduler will gradually reduce the number of diffusion steps for the subsequent tokens.

The results are shown in Table 17. The heuristics lead to comparable performance. For instance, MAR-B with the uncertainty heuristic achieves a $6.0\times$ speed-up while maintaining similar generation quality.

The offline methods perform well overall. In comparison, online methods have the advantage of adjusting the schedule for each sample, which could potentially lead to better results. However, it is challenging to estimate the heuristics accurately and efficiently during generation. One possible direction to explore is introducing momentum to help stabilize the online-calculated heuristics.

### A.2.6 MORE EXAMPLES

We also provide more generated examples. For MAR and Harmon, we find that under the same random seed, the model still generates different styles of images, so we directly show generated examples for MAR and Harmon with DiSA in Figure 8. For FlowAR and xAR, under the same random seed, the model generates similar quality images with and without DiSA, as shown in Figures 9-14.

### A.3 MAR VS. MAE

We show that an MLP can well predict the remaining tokens. This bridges the underlying mechanism between MAR and masked auto-encoder (MAE) (He et al., 2022). Here, we do a preliminary exploration on the connection with MAR and MAE.

MAR-H

Harmon-1.5B

*A realistic landscape shot of the Northern Lights dancing over a snowy mountain range in Iceland.*

*Cute small dog siting in a movie theater eating popcorn watching a movie.*

*Dark high contrast render of a psychedelic tree of life illuminating dust in a mystical cave.*

*A cloud dragon flying over mountains, its body swirling with the wind.*

*A space explorer discovering an alien jungle planet under a purple sky.*

*A close-up photo of a baby sloth holding a treasure chest.*

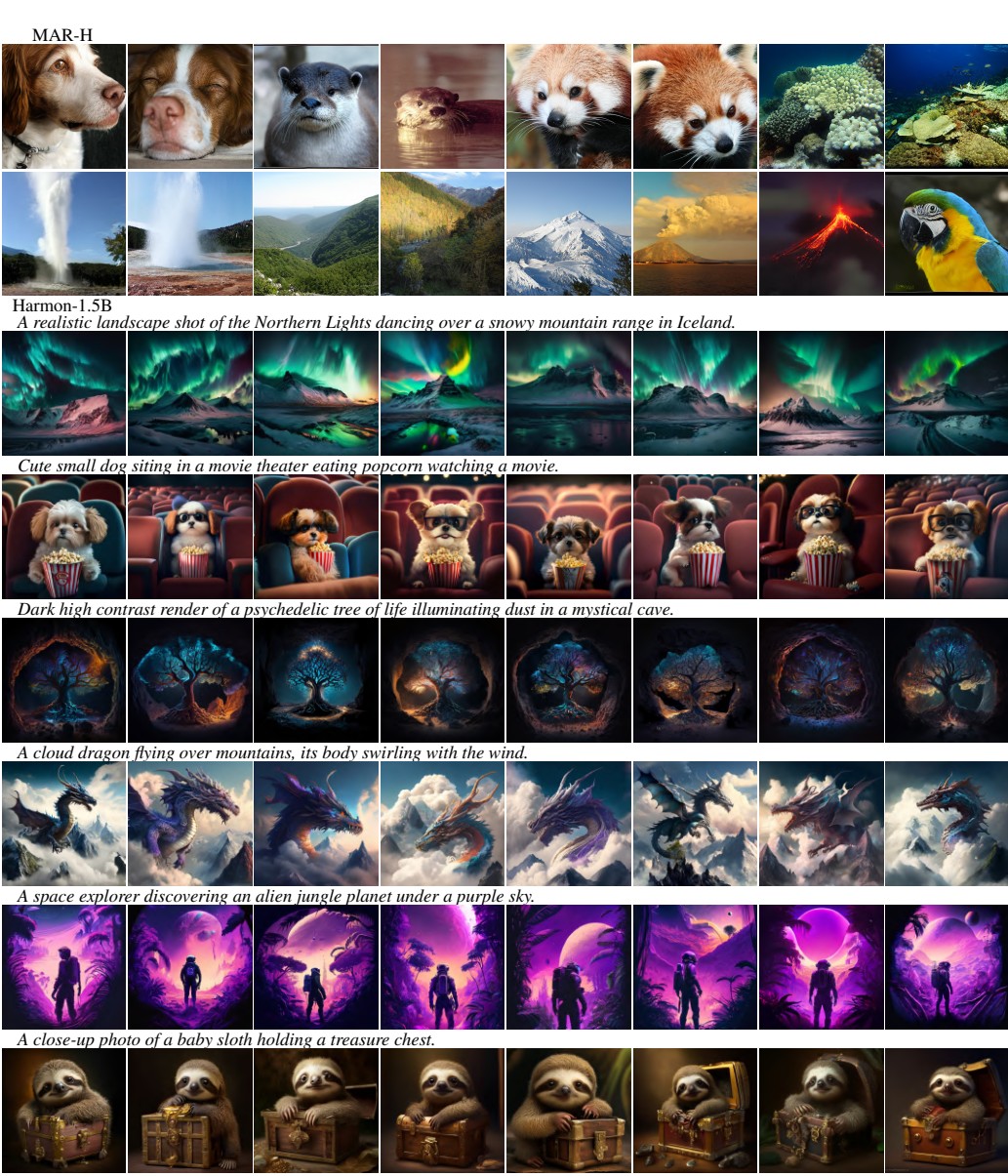

Figure 8: **Generated Samples** from Mar-H and Harmon-1.5B with DiSA.

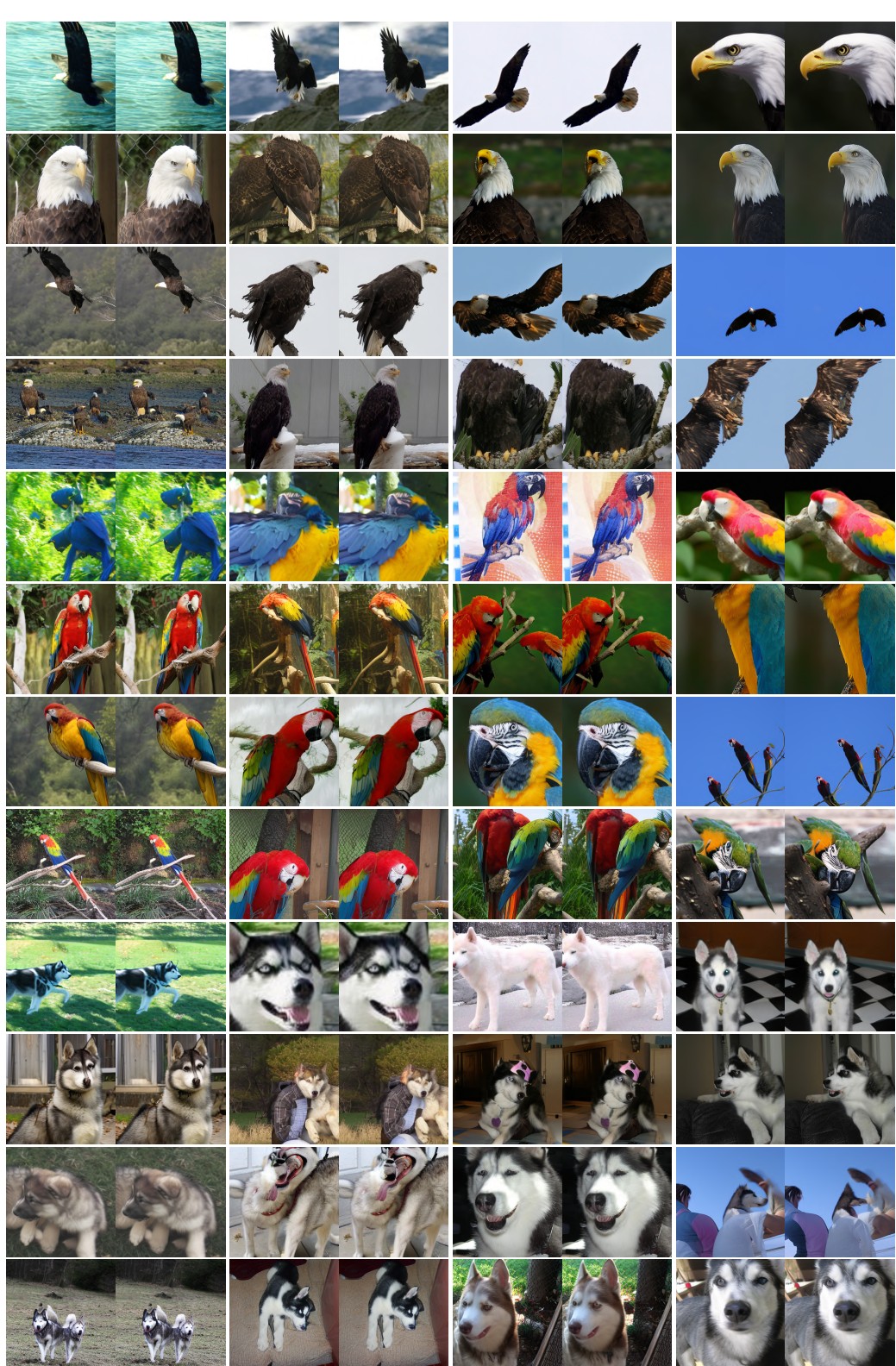

Figure 9: **Generated Samples from FlowAR-H.** Each image pair is generated with the same random seed, where the first is generated without DiSA while the other is with DiSA. We find that DiSA helps generate similar quality images while speeding up image generation by $2.5\times$. Class condition: American eagle (22), macaw (88), Siberian husky (250).

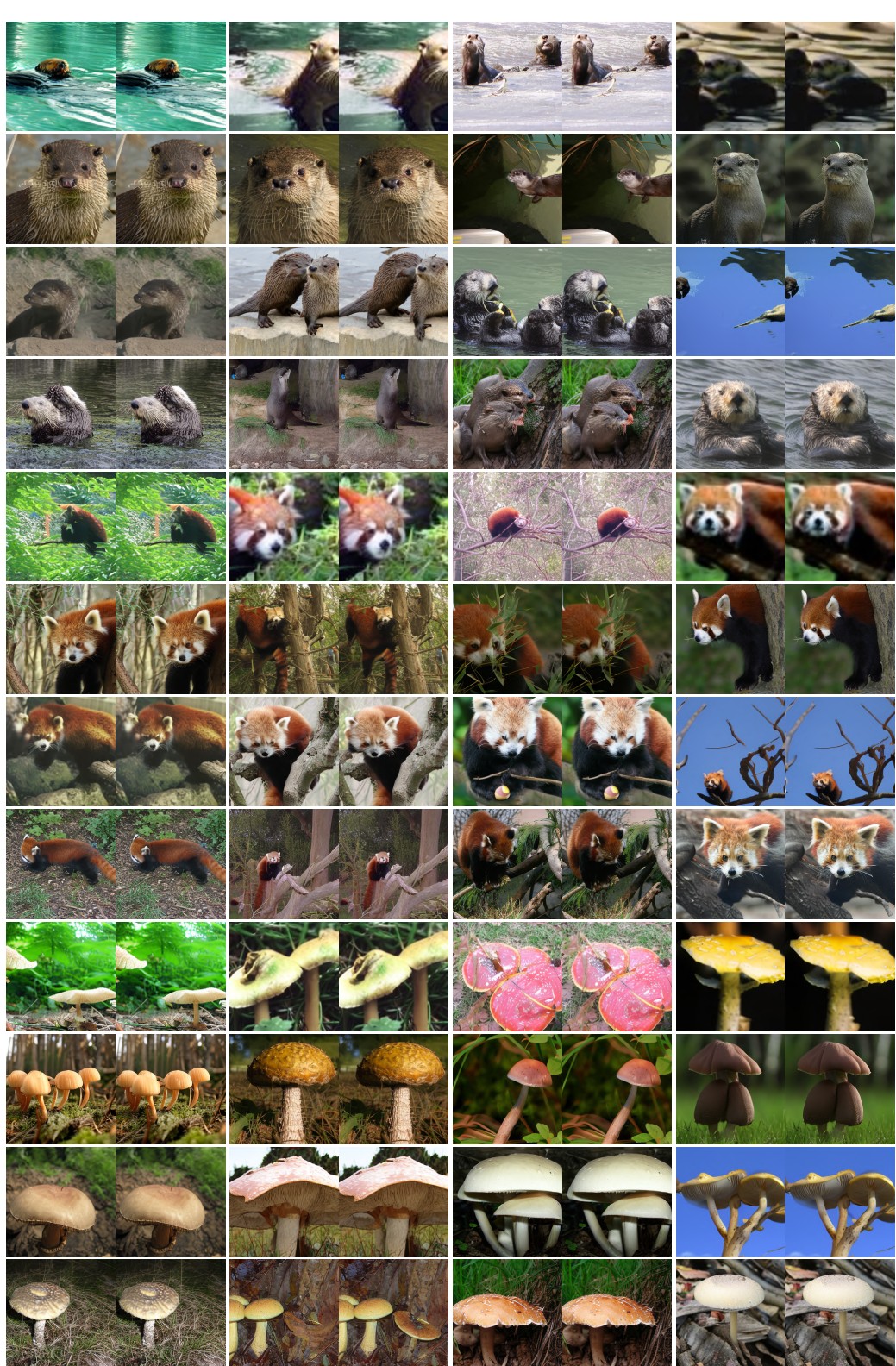

Figure 10: **Generated Samples from FlowAR-H.** Each image pair is generated with the same random seed, where the first is generated without DiSA while the other is with DiSA. We find that DiSA helps generate similar quality images while speeding up image generation by $2.5\times$. Class condition: otter (360), red panda (387), mushroom (947).

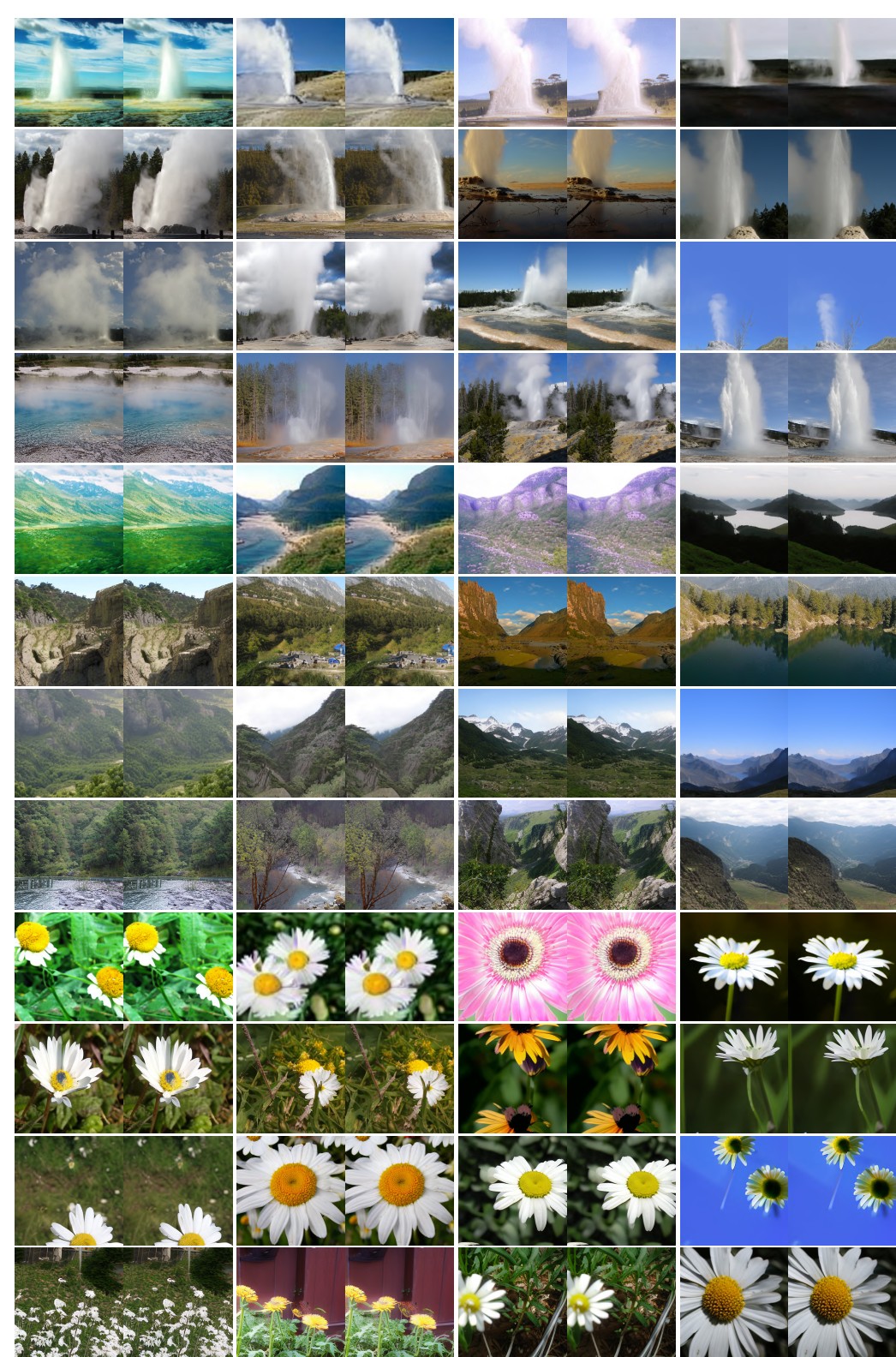

Figure 11: **Generated Samples from FlowAR-H.** Each image pair is generated with the same random seed, where the first is generated without DiSA while the other is with DiSA. We find that DiSA helps generate similar quality images while speeding up image generation by $2.5\times$. Class condition: geyser (974), valley (979), daisy (985).

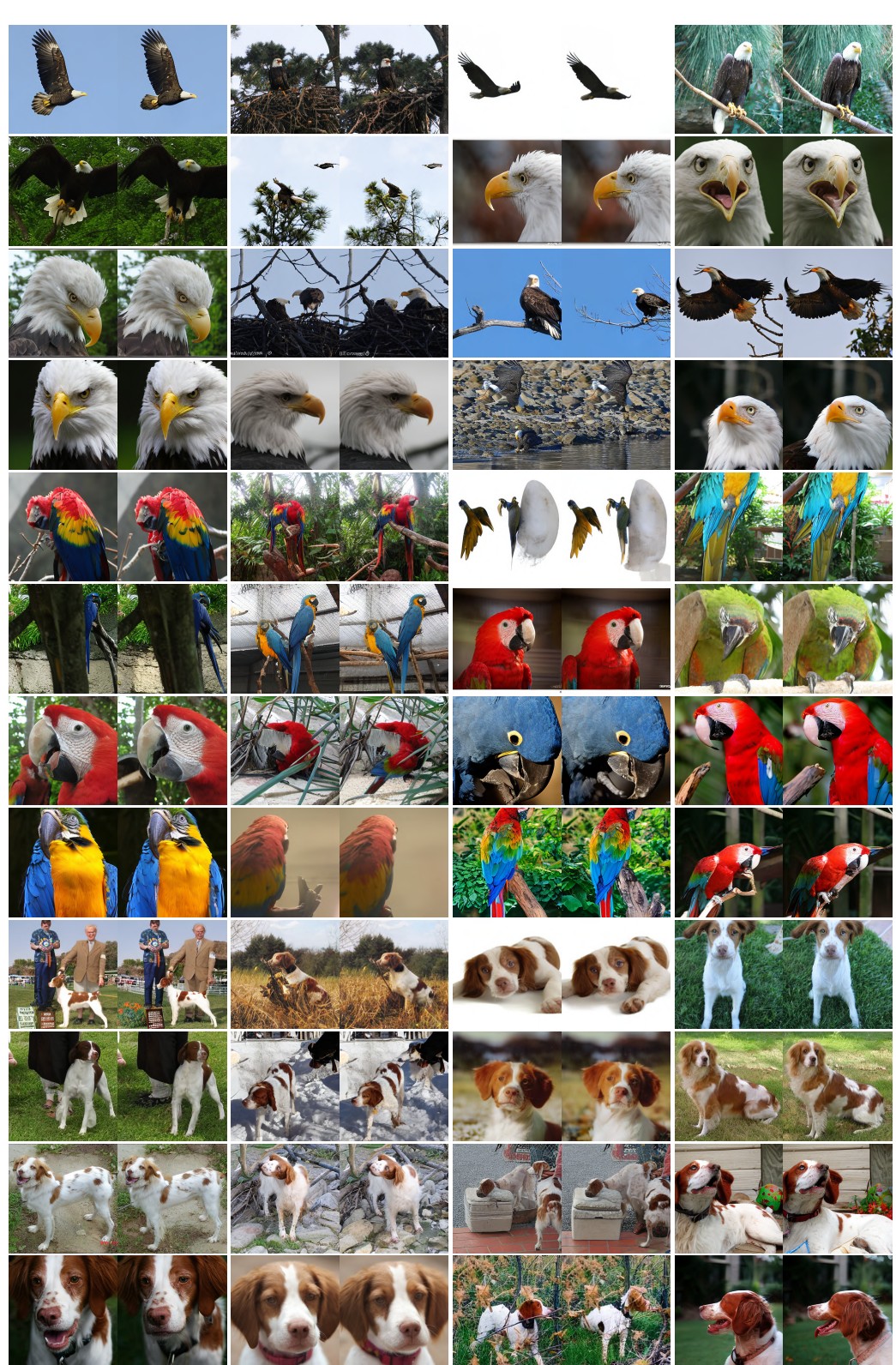

Figure 12: **Generated Samples from xAR-H.** Each image pair is generated with the same random seed, where the first is generated without DiSA while the other is with DiSA. We find that DiSA helps generate similar quality images while speeding up image generation by $1.6\times$. Class condition: American eagle (22), macaw (88), Brittany spaniel (215).

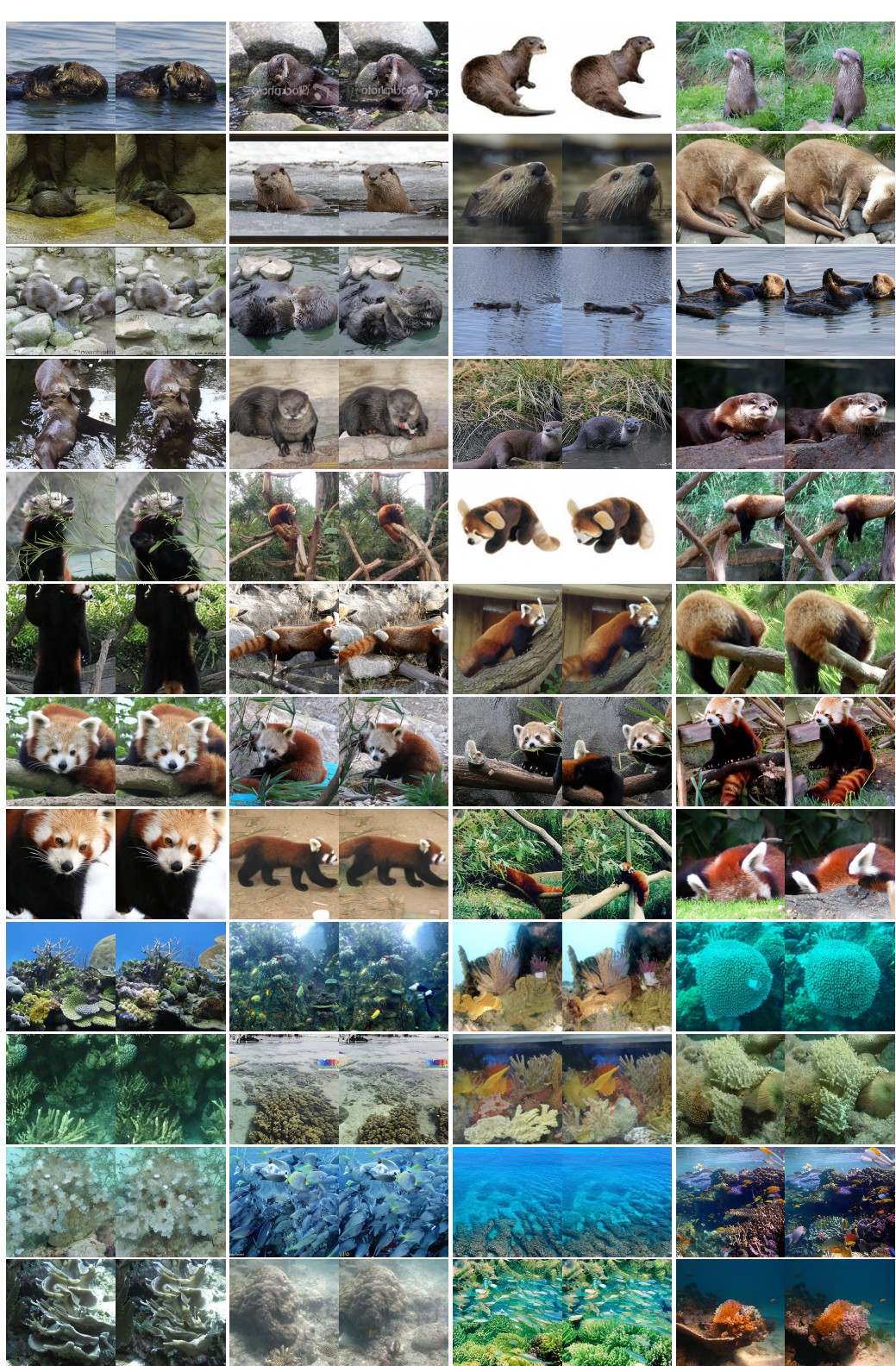

Figure 13: **Generated Samples from xAR-H.** Each image pair is generated with the same random seed, where the first is generated without DiSA while the other is with DiSA. We find that DiSA helps generate similar quality images while speeding up image generation by $1.6\times$. Class condition: otter (360), red panda (387), coral reef (973).

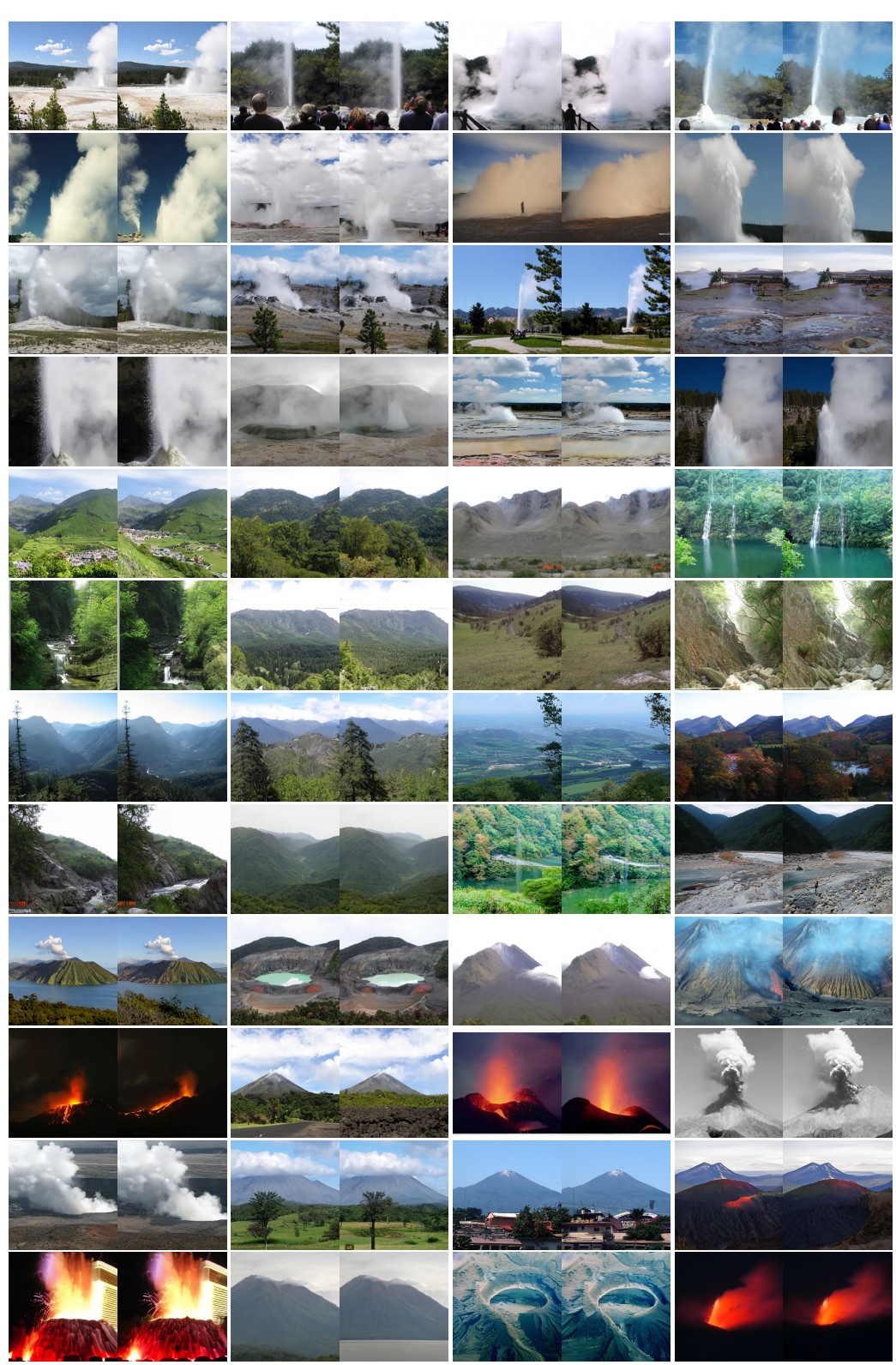

Figure 14: **Generated Samples from xAR-H.** Each image pair is generated with the same random seed, where the first is generated without DiSA while the other is with DiSA. We find that DiSA helps generate similar quality images while speeding up image generation by $1.6\times$. Class condition: geyser (974), valley (979), volcano (980).

Diffusion loss can be written in the equivalent format as follows:

$$\mathcal{L}(z, x) = \mathbb{E}_{\varepsilon,t}\left[\left\|x - \hat{x}_\theta(x_t|t, z)\right\|^2\right] = \mathbb{E}_{\varepsilon,t}\left[\left\|x - \hat{x}_\theta(x_t|t, f(x^1, \ldots, x^{k-1}))\right\|^2\right]. \tag{7}$$

We can also write out the reconstruction loss in Masked Autoencoder (MAE):

$$\mathcal{L}(z, x) = \mathbb{E}_{\varepsilon,t}\left[\left\|x - \hat{x}_\theta(f(x^1, \ldots, x^{k-1}))\right\|^2\right]. \tag{8}$$

Comparing Eq. (7) and Eq. (8), we can see that both loss functions require the model to predict mask tokens. In MAR training, $t$ will sampled from $\{0, 1 \ldots, 999\}$. When $t$ is large, for instance, $t = 950$, $x_t$ is more similar to the Gaussian noise, so it provides little more information than seen tokens. In this case, Eq. (7) actually recovers Eq. (8). Note that, in training iteration, for each token, four different time steps are sampled, the corresponding noisy $x_t$ are generated, and losses on four $x_t$ will be averaged as the loss for one token. And considering the long training process of MAR, $t$ normally will cover the large values.