# OpenReview forum: "DiSA: Diffusion Step Annealing in Autoregressive Image Generation"
_ICLR.cc/2026/Conference — Submitted to ICLR 2026_

### Official Review · Reviewer_phup · 2025-10-25

**Soundness:** 3
**Presentation:** 3
**Contribution:** 3
**Rating:** 6
**Confidence:** 4

**Summary:**

This paper investigates the inefficiency problem in autoregressive image generation models that incorporate diffusion-based sampling (e.g., MAR, FlowAR, xAR, Harmon). These models typically require 50–100 diffusion denoising steps for each token, leading to high inference latency. The paper proposes Diffusion Step Annealing, a training-free inference-time strategy that gradually reduces the number of diffusion steps as more tokens are generated

**Strengths:**

1. The paper presents a convincing empirical study demonstrating that later AR steps have more constrained distributions.
2. Experiment results demonstrate consistent speed-ups across four major AR-diffusion models (MAR, FlowAR, xAR, Harmon) with minimal loss in quality.
3. The paper is well-written, logically structured, concise, and clear, making it easy for readers to understand.

**Weaknesses:**

1. The annealing schedule (linear, cosine, two-stage) and the choice of T_early, T_late are not extensively analyzed. The robustness of these settings across datasets and models could be better demonstrated.
2. The idea of step annealing has precedents in pure diffusion models (e.g., DDIM, DPM-Solver). The novelty here lies in transferring and validating this principle within autoregressive-diffusion frameworks.

**Questions:**

See weaknesses.

---

> ### Author Response · Authors · 2025-11-19
>
> We greatly appreciate your time and effort in reviewing our paper. We are pleased that you acknowledge our paper is well-written, presents convincing empirical study to support out motivation, and shows consistent speed-ups of our method across four major AR-diffusion models.
>
> > **Q1. The annealing schedule (linear, cosine, two-stage) and the choice of T_early, T_late are not extensively analyzed across datasets and models.**
>
> Thank you! As suggested, we conduct a hyperparameter sweep to analyze the performance sensitivity to the annealing schedule.
> The results are shown in the global response. As shown, although the performance degrades under extreme schedules (e.g., AR step=8, Diff steps 50$\rightarrow$5 for MAR-B), DiSA performs robustly across a wide range of schedule designs.
>
> We would like to note that tuning DiSA is very simple. In our experiments, we use the CFG values recommended by the original papers and set $T_{early}$ match the original diffusion steps used in each model. We only need to tune $T_{late}$. The way that we use DiSA is similar to tuning the CFG values — set a value for $T_{late}$ and check the generation quality.
>
> The results of sensitivity analyses are partly presented in Tables 5-14 and partly visualized in Figure 4 in our paper. We will revise the manuscript and add more visualizations to include the new experiments.
>
> Besides, we conduct experiments on the text-to-video benchmark with different metrics when replying to Reviewer 9xYm. We kindly refer you to there for more results.
>
> > **Q2. The idea of step annealing has precedents in pure diffusion models (e.g., DDIM, DPM-Solver). The novelty here lies in transferring and validating this principle within autoregressive-diffusion frameworks.**
>
> Thank you! It is a common misunderstanding that our study is just "transferring" diffusion acceleration techniques to AR-diffusion models. But we highlight that our method is orthogonal to prior diffusion acceleration techniques (e.g., DDIM, DPM-Solver) and is designed specifically for AR-diffusion.
>
> Prior techniques reduce the diffusion steps for the whole image. DDIM achieves a significant speed-up over DDPM via constructing a class of non-Markovian diffusion processes, while DPM-Solver and other advanced solvers use high-order ODE solvers to reduce the numerical error in ODE simulation. Note that these methods decrease the number of denoising steps for each image and all image tokens share the same number of denoising steps.
>
> Our motivation and mechanism are different. First, our key insight is that during AR process, the conditioning for generating new tokens grows stronger. This is different from prior techniques and is specific to AR-diffusion models. Second, our method gradually reduces the diffusion steps in the AR process. For instance, DiSA may use 100 diffusion steps for the first token, but only around 5 steps for the last token. This step annealing happens during the AR process for each image and our design is orthogonal to prior diffusion acceleration methods.
>
> Because our approach is orthogonal to prior methods, it can be combined with DDIM or DPM-Solver. Specifically, we apply DDIM or DPM-Solver to accelerate the diffusion process in MAR and also anneals the diffusion step during the AR process. As shown in Table 3, DiSA can work together with these diffusion-only acceleration techniques on AR-diffusion models and provide further improvements.
>
> -----
> We greatly appreciate your valuable feedback.

---

> > ### Comment · Reviewer_phup · 2025-11-25
> >
> > I find the authors' clarification in Q2 unconvincing regarding the method's novelty. I will be lowering my score.

---

> ### Author Response · Authors · 2025-11-25
>
> Thank you for your feedback! Our previous response is not very clear, and we restate it here in a more structured way. We hope our new explanation is better and can address your concern.
>
> > **Q2. The idea of step annealing has precedents in pure diffusion models (e.g., DDIM, DPM-Solver). The novelty here lies in transferring and validating this principle within autoregressive-diffusion frameworks.**
>
> We would like to emphasize that DiSA is not simply “transferring” diffusion-only acceleration techniques to AR-diffusion models and it is designed specifically for AR-diffusion models.
>
> **Motivation.** Our key motivation comes from an insight specific to AR-diffusion models: during the AR process, the conditioning becomes stronger, so later tokens are increasingly easier to generate than earlier ones. This property is resulted from the AR process. In contrast, methods such as DDIM and DPM-Solver are designed purely for diffusion and do not involve an AR-style generation process. For example, DDIM builds non-Markovian diffusion processes, and DPM-Solver reduces numerical errors in ODE-based sampling. Therefore, our method is motivated by an insight on the AR process,  which is different from prior diffusion-only methods.
>
> **Empirical evidence.** In our work, we also provide three empirical findings to support our insight: mask token prediction, variance of diffusion sampling, and straightness of denoising paths. These results help us better understand the properties of the diffusion process during the AR generation and differ our study from prior diffusion-only techniques such as DDIM or DPM-Solver.
>
> **Mechanism.** Existing acceleration methods apply the same reduced number of diffusion steps to all tokens: the first and the last token share the same diffusion schedule. Therefore, these prior methods actually perform global “step reduction" rather than "step annealing". In contrast, DiSA performs non-uniform step annealing along the AR order: earlier tokens use more diffusion steps, and later tokens use fewer steps. We would like to highlight that this mechanism is unique to AR-diffusion models.
>
> **Orthogonality to prior methods.** Because DiSA is designed from a different perspective on the AR process, it is compatible with DDIM, DPM-Solver, and other diffusion-based accelerations. As shown in Table 3, combining with DiSA also improves the inference speed of diffusion-only methods on MAR while maintaining a comparable generation quality.
>
> ----
> We would be very grateful if you could let us know which parts of our explanation remain unconvincing or why you see DiSA as similar to DDIM or DPM-Solver. We will be glad to address these points more concretely in our next response.

---

### Official Review · Reviewer_XbsH · 2025-11-01

**Soundness:** 3
**Presentation:** 3
**Contribution:** 2
**Rating:** 4
**Confidence:** 2

**Summary:**

The authors show that in AR with diffusion architectures, the later AR steps have tighter token distributions and straighter denoising paths, so diffusion can run with fewer steps without hurting quality. They propose DiSA, a training‑free schedule that uses more diffusion steps early and gradually fewer later (two‑stage / linear / cosine schedulers), reducing per‑token diffusion effort as conditions strengthen. Evidence includes: (i) MLPs can better predict later tokens; (ii) the variance of diffusion‑sampled tokens decreases with AR progress; and (iii) path straightness increases (Fig. 3). Finally, the authors show the effectiveness of DiSA on ImageNet‑256 with various AR models.

**Strengths:**

* Clear empirical insight (straighter late‑stage denoising) turned into a simple, general sampler schedule that’s easy to be equipped with various AR + diffusion architectures (Fig. 1)

* Strong evaluation: various image‑level metrics (such as FID/IS/Precision/Recall), per‑image time, and complements existing diffusion accelerators (Table 3).

* Practical wins on both ImageNet 256 x 256 and T2I GenEval (Harmon) with concrete speed–quality curves (Fig. 5).

**Weaknesses:**

I'm not an expert in this area, but I have some concerns and questions based on my understanding.

* About novelty

I appreciate the practical acceleration idea, but the paper mainly relies on the diffusion-step annealing strategy without much theoretical/mathematical evidence. Even a bit of math or intuition on why this schedule makes sense would make the work solid.

* Scheduler robustness

How sensitive is performance to the exact annealing schedule (e.g., 50 -> 5)? Could the authors provide a hyper‑sweep and an auto‑tuning rule per model?

* Automatic scheduling

The proposed heuristics look promising but ad‑hoc. Can the method learn a schedule online from uncertainty/variance signals?

* About experiments

The experiments appear to focus mainly on 256×256-scale data (largely ImageNet or similar), which raises questions about generalization. It would be helpful to see results on higher-resolution settings (e.g., 512×512, 1024×1024) since frequency characteristics can change substantially with resolution and texture complexity.

**Questions:**

Please see the weaknesses.

---

> ### Author Response · Authors · 2025-11-19
>
> We greatly appreciate your time and effort in reviewing our paper. We are pleased that you acknowledge our method is simple with clear empirical insight and strong experimental results.
>
> > **Q1. Even a bit of math or intuition on why this schedule makes sense would make the work solid.**
>
> Thank you!
>
> **Intuition.** Our key motivation is that as more tokens are generated during the AR process, subsequent tokens follow more constrained distributions and are easier to sample. To intuitively explain, let's imagine that the model is generating an image of a dog. When the model has generated part of a dog, the remaining tokens must complete the dog and thus are more constrained.
>
> We show empirically evidence to support our intuition.
>
> First, we train a multilayer perceptron (MLP) or repurpose the original model head, based on the hidden representation of generated tokens, to predict the outcomes of the diffusion process. As shown in Figure 2, in early stages of generation, the prediction is inaccurate and lacks details. But as more tokens are generated, the prediction becomes increasingly accurate, indicating that the AR model now provides stronger conditions for the diffusion head.
>
> Second, variance in diffusion sampling gradually decreases during generation (Figure 3b), indicating that the distribution of the next token becomes increasingly constrained.
>
> **Mathematical justification.** Liu et.al. [1] proposes that straight paths from noise to data distribution are preferred, because they can be simulated with coarse time discretization and hence need fewer steps at inference time.
>
> Inspired by this, we measure the straightness of denoising paths in each AR step, which is defined as the cosine similarity between the predicted score (the gradient of the data distribution density) and the straight direction from the noisy token to the clean token. Equations are shown in the Appendix (lines 681-701).
>
> As shown in Figure 3c, in the later stage of generation, the diffusion paths in MAR become closer to a straight line, indicating that we can use larger step sizes and fewer diffusion steps.
>
> [1] Flow Straight and Fast: Learning to Generate and Transfer Data with Rectified Flow. ICLR 2023.
>
> > **Q2. How sensitive is performance to the exact annealing schedule (e.g., 50 -> 5)? Could the authors provide a hyper‑sweep?**
>
> Thank you! As suggested, we conduct a hyperparameter sweep to analyze the performance sensitivity to the annealing schedule, shown in the global response above. As shown, although the performance degrades under extreme schedules (e.g., AR step=8, Diff steps 50$\rightarrow$5 for MAR-B), DiSA performs robustly across a wide range of schedule designs.

---

> > ### Author Response · Authors · 2025-11-19
> >
> > > **Q3. Could the authors provide an auto‑tuning rule per model?**
> >
> > Thank you! Automatically adjusting the denoising schedule is an interesting research direction.
> >
> > Based on the three empirical experiments in our paper, we explore three heuristics to guide the scheduler. These heuristics can generate different diffusion step schedules for different models.
> >
> > - Straightness of denoising paths.
> >
> > - Variance of diffusion-sampled tokens.
> >
> > - Uncertainty in predicting mask tokens.
> >
> > There are two ways to use these heuristics:
> >
> > - Offline: We let the model generate 50K images, measure the heuristic values, and then design a fixed schedule based on that.
> >
> > - Online: During generation, the heuristics evaluate the generation process of the current tokens. For instance, if the variance of current tokens keeps decreasing, the scheduler will gradually reduce the number of diffusion steps for the subsequent tokens.
> >
> > The results are shown as follows.
> >
> > | Model | AR steps | Diff steps  | FID$\downarrow$ | IS$\uparrow$ | Time (s)$\downarrow$ | Speed-Up$\uparrow$ |
> > | -- | -- | -- | -- | -- | -- | -- |
> > | MAR-B | 256 | 100 | 2.31 | 281.7 | 0.650 | 1.0$\times$ |
> > | | 64 | 50 | 2.39 (+0.08) | 281.0 (-0.7) | 0.134 | 4.8$\times$ |
> > | +DiSA | 64 | 50$\rightarrow$5 | 2.31 (+0.00) | 282.3 (+0.6) | 0.114 | 5.7$\times$ |
> > | +DiSA (Offline) | 64 | Straightness heuristics | 2.30 (-0.01) | 282.2 (+0.5) | 0.120 | 5.4$\times$ |
> > | | 64 | Variance heuristics | 2.31 (+0.00) | 283.0 (+1.3) | 0.108 | 6.0$\times$ |
> > | | 64 | Uncertainty heuristics | 2.30 (-0.01) | 283.9 (+2.2) | 0.109 | 6.0$\times$ |
> > | +DiSA (Online) | 64 | Straightness heuristics | 2.29 (-0.02) | 279.0 (-2.7) | 0.129 | 5.0$\times$ |
> > | | 64 | Variance heuristics | 2.40 (+0.09) | 281.2 (-0.5) | 0.138 | 4.7$\times$ |
> > | | 64 | Uncertainty heuristics | 2.30 (-0.01) | 282.2 (+0.5) | 0.109 | 6.0$\times$ |
> > | MAR-L | 256 | 100 | 1.78 | 296.0 | 1.102 | 1.0$\times$ |
> > | | 64 | 50 | 1.86 (+0.08) | 294.0 (-2.0) | 0.250 | 4.4$\times$ |
> > | +DiSA | 64 | 50$\rightarrow$5 | 1.77 (-0.01) | 298.3 (+2.3) | 0.216 | 5.1$\times$ |
> > | +DiSA (Offline) | 64 | Straightness heuristics | 1.80 (+0.02) | 292.2 (-3.8) | 0.231 | 4.8$\times$ |
> > | | 64 | Variance heuristics | 1.81 (+0.03) | 292.8 (-3.2) | 0.222 | 5.0$\times$ |
> > | | 64 | Uncertainty heuristics | 1.78 (+0.00) | 295.3 (-0.7) | 0.198 | 5.6$\times$ |
> > | +DiSA (Online) | 64 | Straightness heuristics | 1.82 (+0.04) | 291.4 (-4.6) | 0.245 | 4.5$\times$ |
> > | | 64 | Variance heuristics | 1.81 (+0.03) | 291.6 (-4.4) | 0.261 | 4.2$\times$ |
> > | | 64 | Uncertainty heuristics | 1.78 (+0.00) | 295.2 (-0.8) | 0.198 | 5.6$\times$ |
> > | FlowAR-H | 5 | 50 | 1.67 | 276.3 | 0.423 | 1.0$\times$ |
> > | +DiSA | 5 | 50$\rightarrow$15 | 1.69 (+0.02) | 273.8 (-2.5) | 0.167 | 2.5$\times$ |
> > | +DiSA (Offline) | 5 | Straightness heuristics | 1.70 (+0.03) | 275.3 (-1.0) | 0.189 | 2.2$\times$ |
> > | | 5 | Variance heuristics | 1.71 (+0.04) | 277.1 (+0.8) | 0.196 | 2.2$\times$ |
> > | | 5 | Uncertainty heuristics | 1.87 (+0.20) | 281.8 (+5.5) | 0.159 | 2.7$\times$ |
> > | +DiSA (Online) | 5 | Straightness heuristics | 1.71 (+0.04) | 276.0 (-0.3) | 0.197 | 2.1$\times$ |
> > | | 5 | Variance heuristics | 1.72 (+0.05) | 274.5 (-1.8) | 0.404 | 1.0$\times$ |
> > | | 5 | Uncertainty heuristics | 1.82 (+0.15) | 283.0 (+6.7) | 0.181 | 2.3$\times$ |
> > | xAR-H | 4 | 50 | 1.24 | 301.6 | 0.896 | 1.0$\times$ |
> > | +DiSA | 4 | 50$\rightarrow$15 | 1.23 (-0.01) | 300.5 (-1.1) | 0.577 | 1.6$\times$ |
> > | +DiSA (Offline) | 4 | Straightness heuristics | 1.26 (+0.02) | 298.5 (-3.1) | 0.478 | 1.9$\times$ |
> > | | 4 | Variance heuristics | 1.27 (+0.03) | 299.2 (-2.4) | 0.495 | 1.8$\times$ |
> > | | 4 | Uncertainty heuristics | 1.24 (+0.00) | 298.8 (-2.8) | 0.521 | 1.7$\times$ |
> > | +DiSA (Online) | 4 | Straightness heuristics |  1.25 (+0.01) | 299.9 (-1.7) | 0.611 | 1.5$\times$ |
> > | | 4 | Variance heuristics | - | -  | - | Too slow |
> > | | 4 | Uncertainty heuristics | 1.27 (+0.03) | 299.9 (-1.7) | 0.543 | 1.7$\times$ |
> >
> > As shown, the heuristics lead to comparable performance. For instance, MAR-B with the uncertainty heuristic achieves a $6.0\times$ speed-up while maintaining similar generation quality.
> >
> > > **Q4. Can the method learn a heuristic schedule online from uncertainty/variance signals?**
> >
> > Thank you! We also include online results in the table above. In comparison, online methods have the advantage of adjusting the schedule for each sample, which could potentially lead to better results. However, it is challenging to estimate the heuristics accurately and efficiently during generation. One possible direction to explore is introducing momentum to help stabilize the online-calculated heuristics.

---

> ### Author Response · Authors · 2025-11-19
>
> > **Q5. It would be helpful to see results on higher-resolution settings (e.g., 512×512, 1024×1024)**
>
> Thank you for your suggestion! We extend DiSA to MAR on ImageNet 512x512 generation task and Harmon on 512x512 text-to-image generation. We also find a new AR-diffusion model, NOVA [1] during rebuttal, and successfully apply DiSA to NOVA on 1024x1024 text-to-image generation. Results are shown in the global response above.
>
> As shown in our results, DiSA can scale to higher resolutions such as 512×512 and 1024×1024, achieving great acceleration while preserving generation quality. We note that high-resolution tasks involve significantly more tokens than 256×256 generation (e.g., 1024 tokens for 512×512 versus 256 tokens for 256×256).
>
> [1] NOVA: Autoregressive Video Generation without Vector Quantization. ICLR 2025.
>
> ----
> We deeply appreciate your valuable feedback.

---

### Official Review · Reviewer_9xYm · 2025-11-02

**Soundness:** 3
**Presentation:** 3
**Contribution:** 2
**Rating:** 6
**Confidence:** 4

**Summary:**

This paper proposes a training-free strategy to accelerate AR diffusion models by gradually reducing the number of diffusion steps during token generation. Through empirical analysis, the authors find that as more tokens are generated, the diffusion head becomes increasingly constrained, and later tokens require fewer denoising steps. Based on this observation, DiSA linearly anneals the diffusion steps, achieving notable speed-up while maintaining comparable FID and IS scores. The method is plug-and-play and complementary to existing diffusion samplers.

**Strengths:**

DiSA introduces a new interpretation of diffusion dynamics within AR generation. As conditioning strengthens across timesteps, the diffusion process becomes inherently easier. Unlike prior accelerators (e.g., DDIM, DPM-Solver, LazyMAR), which assume uniform difficulty and globally reduce steps, DiSA models the heterogeneity of diffusion necessity over AR progression. This observation is theoretically supported through denoising-path straightness analysis, linking DiSA to the geometry of diffusion ODEs and providing a principled foundation rather than a heuristic adjustment.

The paper substantiates its hypothesis through three orthogonal metrics (prediction accuracy, variance reduction, and trajectory straightness) forming a multi-faceted empirical argument. This triangulated evidence distinguishes DiSA from earlier works that rely solely on output metrics like FID or IS. The combination of quantitative analysis and visual trajectory interpretation strengthens the empirical credibility of its claims.

DiSA is a training-free and architecture-agnostic plug-in, requiring no parameter updates or structural modifications. By only modifying the diffusion step schedule (e.g., from 50→5), it achieves up to 5–10× acceleration on MAR and 1.4–2.5× on FlowAR/xAR with negligible degradation in generation quality. The efficiency-to-complexity ratio clearly surpasses methods like FAR or speculative decoding, demonstrating elegance through minimal intervention.

**Weaknesses:**

While DiSA is empirically well-justified, it remains largely heuristic. The diffusion-step schedule is fixed (typically linear), without a principled derivation from diffusion dynamics or uncertainty theory. In contrast, prior works like AdaDiff or Rectified Flow introduce adaptive step sizes based on explicit error or confidence estimation. DiSA assumes the AR step index monotonically correlates with conditional strength, an assumption not guaranteed for complex prompts. A theoretical analysis linking token entropy or local curvature to optimal diffusion steps would strengthen generality and interpretability.

The training-free nature is practical but introduces a potential mismatch: DiSA modifies the inference-time denoising schedule without retraining the diffusion head, which was originally optimized for uniform timesteps. This causes instability in some models (e.g., MAR required time-offset corrections). By contrast, retraining-based accelerators (e.g., FAR) maintain consistency between training and inference dynamics. Exploring fine-tuning or joint schedule learning could reduce this gap.

All evaluations are limited to ImageNet-256 and GenEval benchmarks. The method’s behavior under higher resolutions, complex spatial layouts, or multimodal conditioning remains untested. Additionally, the study reports only FID and IS, omitting perceptual (LPIPS), semantic (CLIP-Sim), or human-alignment metrics used in recent acceleration research. This narrower evaluation spectrum makes it difficult to quantify subtle degradation patterns or aesthetic trade-offs.

**Questions:**

The experiments focus on 256×256 ImageNet and GenEval.
Could the authors comment on expected behavior for higher resolutions or long-horizon text prompts? For instance, when later tokens represent finer local details, does the “easier-later” assumption still hold?

DiSA shows minimal quality loss, but where does degradation start?
Please provide an analysis or visualization showing at what step reduction threshold (e.g., 50→1, 50→10) artifacts begin to appear. This would help position DiSA’s safe operating range.

Section 5 briefly mentions heuristics (variance, uncertainty, straightness) for online adjustment.
Could the authors expand on how these heuristics performed and whether they can form the basis of a truly adaptive DiSA variant? This seems like a promising direction that could elevate the contribution beyond a fixed schedule

---

> ### Author Response · Authors · 2025-11-19
>
> We sincerely appreciate your time and effort in reviewing our paper. We are glad that you acknowledge our key motivation is supported theoretically through denoising-path straightness analysis and empirically by multi-faceted experiments. We are also pleased that you find our method is simple and useful in accelerating AR-diffusin models, "demonstrating elegance through minimal intervention".
>
> ### Evaluation
> > **Q1. Please provide an analysis or visualization showing at what step reduction threshold (e.g., 50→1, 50→10) artifacts begin to appear.**
>
> Thank you! As suggested, we conduct a hyperparameter sweep to analyze the performance sensitivity to the annealing schedule.
> The results are shown in the global response. As shown, although the performance degrades under extreme schedules (e.g., AR step=8, Diff steps 50$\rightarrow$5 for MAR-B), DiSA performs robustly across a wide range of schedule designs.
>
> We would like to note that tuning DiSA is very simple. In our experiments, we use the CFG values recommended by the original papers and set $T_{early}$ match the original diffusion steps used in each model. We only need to tune $T_{late}$. The way that we use DiSA is similar to tuning the CFG values — set a value for $T_{late}$ and check the generation quality.
>
> The results of sensitivity analyses are partly presented in Tables 5-14 and partly visualized in Figure 4 in our paper. We will revise the manuscript and add more visualizations to include the new experiments.
>
> > **Q2. Experiments on higher resolutions or long-horizon text prompts.**
>
> We extend DiSA to MAR on ImageNet 512x512 generation task and Harmon on 512x512 text-to-image generation. We also find a new AR-diffusion model, NOVA [1] during rebuttal, and successfully apply DiSA to NOVA on 1024x1024 text-to-image generation. Results are shown in the global response.
>
> As shown in our results, DiSA can scale to higher resolutions such as 512×512 and 1024×1024, achieving great acceleration while preserving generation quality. We note that high-resolution tasks involve significantly more tokens than 256×256 generation (e.g., 1024 tokens for 512×512 versus 256 tokens for 256×256).
>
> In our experiments, the text-to-image evaluation is conducted on the GenEval benchmark, where the prompts are not long-horizon, as you pointed out. Nevertheless, we believe DiSA should also work with long-horizon prompts, as our core motivation—that conditioning becomes stronger across AR steps—still holds in such settings. We will further explore this by finding a suitable benchmark with longer and more complex prompts.
>
> [1] NOVA: Autoregressive Video Generation without Vector Quantization. ICLR 2025.
>
> > **Q3. The study reports only FID and IS, omitting perceptual (LPIPS), semantic (CLIP-Sim), or human-alignment metrics used in recent acceleration research.**
>
> We apply DiSA to NOVA on the text-to-video generation task. The evaluation is based on VBench [2], which comprises 16 dimensions in video generation including styles, subject identity inconsistency, motion smoothness, temporal flickering, spatial relationship and so on. Most metrics rely on pretrained models such as CLIP as you suggested and thus capture semantics. Besides, the authors of VBench report scores across all dimensions closely match human perceptions.
>
> ### Overall scores
> | Diff steps | Quality | Semantic | Total Score | Time per video (s) |
> | - | - | - | - | - |
> | 25 | 79.16 | 74.82 | 78.29 | 61 |
> | (+DiSA) 25$\rightarrow$20 | 79.09 | 74.84 | 78.24 | 56 |
> | 25$\rightarrow$15 | 79.23 | 74.84 | 78.35 | 51 |
> | 25$\rightarrow$10 | 79.16 | 74.12 | 78.15 | 46 |
> | 25$\rightarrow$5 | 79.07 | 73.38 | 77.93 | 42 |
>
>
> ### Score of each dimension
>
> | Diff steps |  Subject consistency | Background consistency | Aesthetic quality | Imaging quality | Object class | Multiple objects | Color | Spatial relationship | Scene | Temporal style | Overall consistency | Human action | Temporal flickering | Motion smoothness | Dynamic degree | Appearance style |
> | - | - | - | - | - | - | - | - | - | - | - | - | - | - | - | - | - |
> | 25 | 92.87 | 94.71 | 54.90 | 49.57 | 91.38 | 70.21 | 85.02 | 62.02 | 47.24 | 24.82 | 26.40 | 90.00 | 98.70 | 99.28 | 61.39 | 21.89 |
> | (+DiSA) 25$\rightarrow$20 | 92.35 | 94.56 | 54.94 | 49.23 | 88.18 | 69.09 | 87.40 | 64.03 | 45.68 | 24.84 | 26.67 | 90.80 | 98.65 | 99.23 | 63.33 | 22.00 |
> | 25$\rightarrow$15 | 91.95 | 94.74 | 54.24 | 48.56 | 89.65 | 68.64 | 84.69 | 64.99 | 46.79 | 24.85 | 26.46 | 90.60 | 98.65 | 99.14 | 68.89 | 22.05 |
> | 25$\rightarrow$10 | 91.90 | 94.45 | 54.22 | 47.92 | 89.10 | 66.59 | 86.39 | 63.17 | 45.74 | 24.61 | 26.19 | 90.00 | 98.61 | 99.25 | 69.72 | 21.89 |
> | 25$\rightarrow$5 | 91.88 | 94.22 | 54.22 | 47.52 | 87.99 | 66.27 | 84.77 | 61.81 | 43.74 | 24.59 | 26.12 | 90.60 | 98.61 | 99.17 | 70.56 | 21.86 |
>
> As seen, DiSA also works on NOVA with minimal performance degradation.
>
> [2] VBench: Comprehensive Benchmark Suite for Video Generative Models. CVPR 2024.

---

> ### Author Response · Authors · 2025-11-19
>
> ### Extension to DiSA
> > **Q4. Adaptive DiSA variant with heuristics.**
>
> Thank you for the insightful suggestion! Automatically adjusting the denoising schedule is an interesting research direction.
>
> Based on the three empirical experiments in our paper, we explore three heuristics to guide the scheduler.
>
> - Straightness of denoising paths.
>
> - Variance of diffusion-sampled tokens.
>
> - Uncertainty in predicting mask tokens.
>
> There are two ways to use these heuristics:
>
> - Offline: We let the model generate 50K images, measure the heuristic values, and then design a fixed schedule based on that.
>
> - Online: During generation, the heuristics evaluate the generation process of the current tokens. For instance, if the variance of current tokens keeps decreasing, the scheduler will gradually reduce the number of diffusion steps for the subsequent tokens.
>
> The results are shown as follows.
>
> | Model | AR steps | Diff steps  | FID$\downarrow$ | IS$\uparrow$ | Time (s)$\downarrow$ | Speed-Up$\uparrow$ |
> | -- | -- | -- | -- | -- | -- | -- |
> | MAR-B | 256 | 100 | 2.31 | 281.7 | 0.650 | 1.0$\times$ |
> | | 64 | 50 | 2.39 (+0.08) | 281.0 (-0.7) | 0.134 | 4.8$\times$ |
> | +DiSA | 64 | 50$\rightarrow$5 | 2.31 (+0.00) | 282.3 (+0.6) | 0.114 | 5.7$\times$ |
> | +DiSA (Offline) | 64 | Straightness heuristics | 2.30 (-0.01) | 282.2 (+0.5) | 0.120 | 5.4$\times$ |
> | | 64 | Variance heuristics | 2.31 (+0.00) | 283.0 (+1.3) | 0.108 | 6.0$\times$ |
> | | 64 | Uncertainty heuristics | 2.30 (-0.01) | 283.9 (+2.2) | 0.109 | 6.0$\times$ |
> | +DiSA (Online) | 64 | Straightness heuristics | 2.29 (-0.02) | 279.0 (-2.7) | 0.129 | 5.0$\times$ |
> | | 64 | Variance heuristics | 2.40 (+0.09) | 281.2 (-0.5) | 0.138 | 4.7$\times$ |
> | | 64 | Uncertainty heuristics | 2.30 (-0.01) | 282.2 (+0.5) | 0.109 | 6.0$\times$ |
> | MAR-L | 256 | 100 | 1.78 | 296.0 | 1.102 | 1.0$\times$ |
> | | 64 | 50 | 1.86 (+0.08) | 294.0 (-2.0) | 0.250 | 4.4$\times$ |
> | +DiSA | 64 | 50$\rightarrow$5 | 1.77 (-0.01) | 298.3 (+2.3) | 0.216 | 5.1$\times$ |
> | +DiSA (Offline) | 64 | Straightness heuristics | 1.80 (+0.02) | 292.2 (-3.8) | 0.231 | 4.8$\times$ |
> | | 64 | Variance heuristics | 1.81 (+0.03) | 292.8 (-3.2) | 0.222 | 5.0$\times$ |
> | | 64 | Uncertainty heuristics | 1.78 (+0.00) | 295.3 (-0.7) | 0.198 | 5.6$\times$ |
> | +DiSA (Online) | 64 | Straightness heuristics | 1.82 (+0.04) | 291.4 (-4.6) | 0.245 | 4.5$\times$ |
> | | 64 | Variance heuristics | 1.81 (+0.03) | 291.6 (-4.4) | 0.261 | 4.2$\times$ |
> | | 64 | Uncertainty heuristics | 1.78 (+0.00) | 295.2 (-0.8) | 0.198 | 5.6$\times$ |
> | FlowAR-H | 5 | 50 | 1.67 | 276.3 | 0.423 | 1.0$\times$ |
> | +DiSA | 5 | 50$\rightarrow$15 | 1.69 (+0.02) | 273.8 (-2.5) | 0.167 | 2.5$\times$ |
> | +DiSA (Offline) | 5 | Straightness heuristics | 1.70 (+0.03) | 275.3 (-1.0) | 0.189 | 2.2$\times$ |
> | | 5 | Variance heuristics | 1.71 (+0.04) | 277.1 (+0.8) | 0.196 | 2.2$\times$ |
> | | 5 | Uncertainty heuristics | 1.87 (+0.20) | 281.8 (+5.5) | 0.159 | 2.7$\times$ |
> | +DiSA (Online) | 5 | Straightness heuristics | 1.71 (+0.04) | 276.0 (-0.3) | 0.197 | 2.1$\times$ |
> | | 5 | Variance heuristics | 1.72 (+0.05) | 274.5 (-1.8) | 0.404 | 1.0$\times$ |
> | | 5 | Uncertainty heuristics | 1.82 (+0.15) | 283.0 (+6.7) | 0.181 | 2.3$\times$ |
> | xAR-H | 4 | 50 | 1.24 | 301.6 | 0.896 | 1.0$\times$ |
> | +DiSA | 4 | 50$\rightarrow$15 | 1.23 (-0.01) | 300.5 (-1.1) | 0.577 | 1.6$\times$ |
> | +DiSA (Offline) | 4 | Straightness heuristics | 1.26 (+0.02) | 298.5 (-3.1) | 0.478 | 1.9$\times$ |
> | | 4 | Variance heuristics | 1.27 (+0.03) | 299.2 (-2.4) | 0.495 | 1.8$\times$ |
> | | 4 | Uncertainty heuristics | 1.24 (+0.00) | 298.8 (-2.8) | 0.521 | 1.7$\times$ |
> | +DiSA (Online) | 4 | Straightness heuristics |  1.25 (+0.01) | 299.9 (-1.7) | 0.611 | 1.5$\times$ |
> | | 4 | Variance heuristics | - | -  | - | Too slow |
> | | 4 | Uncertainty heuristics | 1.27 (+0.03) | 299.9 (-1.7) | 0.543 | 1.7$\times$ |
>
> As shown, the heuristics lead to comparable performance. For instance, MAR-B with the uncertainty heuristic achieves a $6.0\times$ speed-up while maintaining similar generation quality. In comparison, online methods have the advantage of adjusting the schedule for each sample, which could potentially lead to better results. However, it is challenging to estimate the heuristics accurately and efficiently during generation. One possible direction to explore is introducing momentum to help stabilize the online-calculated heuristics.
>
> Unfortunately, we acknowledge that the heuristics cannot achieve a fully autonomous version of DiSA because they still require manual design. As noted, tuning DiSA on a given model is very simple, so these heuristics may make the pipeline complex with minimal gains. Thus, we leave the development of a more automated variant as an interesting direction for future work.
>
> We will update our paper accordingly to include this discussion.

---

> ### Author Response · Authors · 2025-11-19
>
> > **Q5. Exploring fine-tuning or joint schedule learning to reduce the train-test gap.**
>
> We strongly agree with you that there is a train-test gap in diffusion models. During training, the timesteps $t$ are typically sampled from {0, ..., 1000} or from a continuous interval such as [0, 1]. In contrast, inference relies on a much smaller subset of timesteps (e.g., only 50 steps). Ideally, a well-trained diffusion model should perform robustly across all timesteps, enabling flexible sampling schedules at inference. Thus, DiSA and prior techniques such as Improved DDPM, DDIM, and DPM-Solver only modifies the inference-time schedule and do not reduce the timestep set at training.
>
> As a preliminary exploration, we train and compare two MAR-Base models from scratch for 200K iterations:
> (1) a model trained with {0, ..., 1000} timesteps while using only a small subset of steps at inference;
> (2) a model whose training and inference share the same schedule (i.e., using only 50 steps).
>
> | Model | FID$\downarrow$ | IS$\uparrow$ |
> | - | - | - |
> | (1) | 31.1 | 46.7 |
> | (2) | 30.1 | 50.0 |
>
> As shown, the model without a train–test gap converges faster, as it is optimized under the same schedule used during inference. However, we do not claim this observation will necessarily hold at larger training scales, and we leave a more systematic study as future work.
>
> > **Q6. A theoretical analysis linking token entropy or local curvature to optimal diffusion steps would strengthen generality and interpretability.**
>
> We appreciate your insightful suggestion. In the paper, we provide a theoretical analysis based on denoising-path straightness, supporting our central claim that conditioning becomes stronger across AR steps. However, determining the theoretically optimal number of diffusion steps is much more challenging. It may depend on the model architecture, the choice of diffusion or flow-matching objectives, and the scheduling strategy. We fully agree that a theory for optimal diffusion steps would further enhance our method, and we view this as an important direction for future work.
>
> ---
> Again, we sincerely appreciate your valuable feedback.

---

### Author Response · Authors · 2025-11-19
**Global response to reviewers: robustness and scalability**

We sincerely thank the reviewers for their insightful and constructive feedback. The reviewers commonly raised two primary concerns regarding DiSA: (1) Sensitivity of DiSA to hyperparameters, including the annealing schedule (linear, cosine, two-stage) and the choice of $T_{early}$ and $T_{late}$; (2) Extension to high-resolution generation. New experiments are detailed in this response to address the two concerns.

> **Sensitivity of DiSA to hyperparameters**

### MAR-B + DiSA (linear)

| Model | AR steps | Diff steps  | FID$\downarrow$ | IS$\uparrow$ | Time (s)$\downarrow$ | Speed-Up$\uparrow$ |
| - | - | - | - | - | - | - |
| MAR-B | 256 | 100 | 2.31 | 281.7 | 0.650 | 1.0$\times$ |
| | 64 | 50 | 2.39 (+0.08) | 281.0 (-0.7) | 0.134 | 4.8$\times$ |
| +DiSA | 64 | 50$\rightarrow$25 | 2.31 (+0.00) | 278.9 (-2.8) | 0.126 | 5.2$\times$ |
|  | 64 | 50$\rightarrow$15 | 2.26 (-0.05) | 281.0 (-0.7) | 0.120 | 5.4$\times$ |
|  | 64 | 50$\rightarrow$10 | 2.29 (-0.02) | 279.3 (-2.4) | 0.119 | 5.5$\times$ |
|  | 64 | 50$\rightarrow$5 | 2.31 (+0.00) | 282.3 (+0.6) | 0.114 | 5.7$\times$ |
|  | 64 | 25$\rightarrow$15 | 2.31 (+0.00) | 280.6 (-1.1) | 0.097 | 6.7$\times$ |
|  | 64 | 25$\rightarrow$10 | 2.38 (+0.07) | 283.6 (+1.9) | 0.094 | 6.9$\times$ |
|  | 64 | 25$\rightarrow$5 | 2.38 (+0.07) | 283.3 (+1.6) | 0.090 | 7.2$\times$ |
|  | 64 | 15$\rightarrow$10 | 2.53 (+0.22) | 287.9 (+6.2) | 0.088 | 7.4$\times$ |
|  | 64 | 15$\rightarrow$5 | 2.86 (+0.55) | 283.9 (+2.2) | 0.084 | 7.7$\times$ |
| - | - | - | - | - | - | - |
|  | 32 | 50$\rightarrow$25 | 2.56 (+0.25) | 272.0 (-9.7) | 0.071 | 9.2$\times$ |
|  | 32 | 50$\rightarrow$15 | 2.47 (+0.16) | 272.0 (-9.7) | 0.067 | 9.7$\times$ |
|  | 32 | 50$\rightarrow$10 | 2.42 (+0.11) | 273.9 (-7.8) | 0.065 | 10.0$\times$ |
|  | 32 | 50$\rightarrow$5 | 2.32 (+0.01) | 279.3 (-2.4) | 0.063 | 10.4$\times$ |
|  | 32 | 25$\rightarrow$15 | 2.45 (+0.14) | 276.3 (-5.4) | 0.053 | 12.3$\times$ |
|  | 32 | 25$\rightarrow$10 | 2.39 (+0.08) | 279.8 (-1.9) | 0.051 | 12.8$\times$ |
|  | 32 | 25$\rightarrow$5 | 2.34 (+0.03) | 285.6 (+3.9) | 0.049 | 13.3$\times$ |
|  | 32 | 15$\rightarrow$10 | 2.44 (+0.13) | 284.3 (+2.6) | 0.048 | 13.6$\times$ |
|  | 32 | 15$\rightarrow$5 | 2.63 (+0.32) | 286.7 (+5.0) | 0.046 | 14.1$\times$ |
| - | - | - | - | - | - | - |
|  | 16 | 50$\rightarrow$25 | 4.33 (+2.02) | 246.4 (-35.3) | 0.045 | 14.4$\times$ |
|  | 16 | 50$\rightarrow$15 | 4.15 (+1.84) | 247.8 (-33.9) | 0.042 | 15.6$\times$ |
|  | 16 | 50$\rightarrow$10 | 4.01 (+1.70) | 249.5 (-32.2) | 0.040 | 16.3$\times$ |
|  | 16 | 50$\rightarrow$5 | 3.65 (+1.34) | 255.2 (-26.5) | 0.038 | 16.9$\times$ |
| - | - | - | - | - | - | - |
|  | 8 | 50$\rightarrow$25 | 13.59 (+11.28) | 179.6 (-102.1) | 0.031 | 20.7$\times$ |
|  | 8 | 50$\rightarrow$15 | 13.15 (+10.84) | 182.9 (-98.8) | 0.029 | 22.5$\times$ |
|  | 8 | 50$\rightarrow$10 | 12.62 (+10.31) | 185.8 (-95.9) | 0.027 | 23.8$\times$ |
|  | 8 | 50$\rightarrow$5 | 10.97 (+8.66) | 196.2 (-85.5) | 0.026 | 25.5$\times$ |

### MAR-B + DiSA (cosine)

| Model | AR steps | Diff steps  | FID$\downarrow$ | IS$\uparrow$ | Time (s)$\downarrow$ | Speed-Up$\uparrow$ |
| -- | -- | -- | -- | -- | -- | -- |
| MAR-B | 256 | 100 | 2.31 | 281.7 | 0.650 | 1.0$\times$ |
| | 64 | 50 | 2.39 (+0.08) | 281.0 (-0.7) | 0.134 | 4.8$\times$ |
| +DiSA | 64 | 50$\rightarrow$25 | 2.31 (+0.00) | 279.3 (-2.4) | 0.117 | 5.5$\times$ |
|  | 64 | 50$\rightarrow$15 | 2.32 (+0.01) | 279.4 (-2.3) | 0.112 | 5.8$\times$ |
|  | 64 | 50$\rightarrow$10 | 2.30 (-0.01) | 280.1 (-1.6) | 0.109 | 5.9$\times$ |
|  | 64 | 50$\rightarrow$5 | 2.33 (+0.02) | 281.3 (-0.4) | 0.105 | 6.2$\times$ |
| - | - | - | - | - | - | - |
|  | 32 | 50$\rightarrow$25 | 2.53 (+0.22) | 273.0 (-8.7) | 0.065 | 10.0$\times$ |
|  | 32 | 50$\rightarrow$15 | 2.44 (+0.13) | 273.0 (-8.7) | 0.061 | 10.7$\times$ |
|  | 32 | 50$\rightarrow$10 | 2.40 (+0.09) | 277.6 (-4.1) | 0.060 | 10.9$\times$ |
|  | 32 | 50$\rightarrow$5 | 2.35 (+0.04) | 277.4 (-4.3) | 0.057 | 11.4$\times$ |
| - | - | - | - | - | - | - |
|  | 16 | 50$\rightarrow$25 | 4.30 (+1.99) | 244.0 (-37.7) | 0.040 | 16.2$\times$ |
|  | 16 | 50$\rightarrow$15 | 4.04 (+1.73) | 248.7 (-33.0) | 0.038 | 17.3$\times$ |
|  | 16 | 50$\rightarrow$10 | 3.81 (+1.50) | 251.9 (-29.8) | 0.036 | 18.3$\times$ |
|  | 16 | 50$\rightarrow$5 | 3.31 (+1.00) | 260.2 (-21.5) | 0.034 | 19.1$\times$ |
| - | - | - | - | - | - | - |
|  | 8 | 50$\rightarrow$25 | 13.43 (+11.12) | 179.2 (-102.5) | 0.028 | 23.0$\times$ |
|  | 8 | 50$\rightarrow$15 | 12.98 (+10.67) | 182.5 (-99.2) | 0.026 | 25.4$\times$ |
|  | 8 | 50$\rightarrow$10 | 12.15 (+9.84) | 188.5 (-93.2) | 0.024 | 26.9$\times$ |
|  | 8 | 50$\rightarrow$5 | 10.09 (+7.78) | 201.2 (-80.5) | 0.023 | 27.9$\times$ |

---

> ### Author Response · Authors · 2025-11-19
>
> ### MAR-B + DiSA (two-stage)
>
> | Model | AR steps | Diff steps  | FID$\downarrow$ | IS$\uparrow$ | Time (s)$\downarrow$ | Speed-Up$\uparrow$ |
> | -- | -- | -- | -- | -- | -- | -- |
> | MAR-B | 256 | 100 | 2.31 | 281.7 | 0.650 | 1.0$\times$ |
> | | 64 | 50 | 2.39 (+0.08) | 281.0 (-0.7) | 0.134 | 4.8$\times$ |
> | +DiSA | 64 | 50$\rightarrow$25 | 2.35 (+0.04) | 279.6 (-2.1) | 0.118 | 5.5$\times$ |
> |  | 64 | 50$\rightarrow$15 | 2.29 (-0.02) | 283.1 (+1.4) | 0.112 | 5.8$\times$ |
> |  | 64 | 50$\rightarrow$10 | 2.37 (+0.06) | 280.4 (-1.3) | 0.111 | 5.9$\times$ |
> |  | 64 | 50$\rightarrow$5 | 3.17 (+0.86) | 270.8 (-10.9) | 0.106 | 6.2$\times$ |
> | - | - | - | - | - | - | - |
> |  | 32 | 50$\rightarrow$25 | 2.52 (+0.21) | 274.3 (-7.4) | 0.065 | 10.1$\times$ |
> |  | 32 | 50$\rightarrow$15 | 2.43 (+0.12) | 274.9 (-6.8) | 0.061 | 10.7$\times$ |
> |  | 32 | 50$\rightarrow$10 | 2.34 (+0.03) | 278.2 (-3.5) | 0.058 | 11.1$\times$ |
> |  | 32 | 50$\rightarrow$5 | 2.87 (+0.56) | 275.9 (-5.8) | 0.057 | 11.4$\times$ |
> | - | - | - | - | - | - | - |
> |  | 16 | 50$\rightarrow$25 | 4.28 (+1.97) | 244.2 (-37.5) | 0.040 | 16.3$\times$ |
> |  | 16 | 50$\rightarrow$15 | 3.88 (+1.57) | 251.8 (-29.9) | 0.037 | 17.5$\times$ |
> |  | 16 | 50$\rightarrow$10 | 3.46 (+1.15) | 255.7 (-26.0) | 0.035 | 18.5$\times$ |
> |  | 16 | 50$\rightarrow$5 | 3.07 (+0.76) | 263.4 (-18.3) | 0.033 | 19.4$\times$ |
> | - | - | - | - | - | - | - |
> |  | 8 | 50$\rightarrow$25 | 13.57 (+11.26) | 180.5 (-101.2) | 0.028 | 23.2$\times$ |
> |  | 8 | 50$\rightarrow$15 | 12.67 (+10.36) | 185.9 (-95.8) | 0.025 | 25.6$\times$ |
> |  | 8 | 50$\rightarrow$10 | 10.90 (+8.59) | 195.6 (-86.1) | 0.024 | 27.1$\times$ |
> |  | 8 | 50$\rightarrow$5 | 7.21 (+4.90) | 219.6 (-62.1) | 0.022 | 28.9$\times$ |
>
> ### MAR-L + DiSA (linear)
> | Model | AR steps | Diff steps  | FID$\downarrow$ | IS$\uparrow$ | Time (s)$\downarrow$ | Speed-Up$\uparrow$ |
> | -- | -- | -- | -- | -- | -- | -- |
> | MAR-L | 256 | 100 | 1.78 | 296.0 | 1.102 | 1.0$\times$ |
> | | 64 | 50 | 1.86 (+0.08) | 294.0 (-2.0) | 0.250 | 4.4$\times$ |
> | +DiSA | 64 | 50$\rightarrow$25 | 1.83 (+0.05) | 290.1 (-5.9) | 0.232 | 4.8$\times$ |
> |  | 64 | 50$\rightarrow$15 | 1.78 (+0.00) | 293.3 (-2.7) | 0.224 | 4.9$\times$ |
> |  | 64 | 50$\rightarrow$10 | 1.80 (+0.02) | 292.3 (-3.7) | 0.225 | 4.9$\times$ |
> |  | 64 | 50$\rightarrow$5 | 1.77 (-0.01) | 298.3 (+2.3) | 0.216 | 5.1$\times$ |
> |  | 64 | 25$\rightarrow$15 | 1.80 (+0.02) | 294.4 (-1.6) | 0.199 | 5.5$\times$ |
> |  | 64 | 25$\rightarrow$10 | 1.82 (+0.04) | 296.4 (+0.4) | 0.196 | 5.6$\times$ |
> |  | 64 | 25$\rightarrow$5 | 1.85 (+0.07) | 298.4 (+2.4) | 0.191 | 5.8$\times$ |
> |  | 64 | 15$\rightarrow$10 | 1.93 (+0.15) | 302.9 (+6.9) | 0.187 | 5.9$\times$ |
> |  | 64 | 15$\rightarrow$5 | 2.26 (+0.48) | 302.1 (+6.1) | 0.183 | 6.0$\times$ |
> | - | - | - | - | - | - | - |
> |  | 32 | 50$\rightarrow$25 | 2.21 (+0.43) | 278.6 (-17.4) | 0.129 | 8.5$\times$ |
> |  | 32 | 50$\rightarrow$15 | 2.14 (+0.36) | 281.6 (-14.4) | 0.123 | 9.0$\times$ |
> |  | 32 | 50$\rightarrow$10 | 2.06 (+0.28) | 281.2 (-14.8) | 0.120 | 9.2$\times$ |
> |  | 32 | 50$\rightarrow$5 | 1.92 (+0.14) | 285.5 (-10.5) | 0.117 | 9.4$\times$ |
> |  | 32 | 25$\rightarrow$15 | 2.07 (+0.29) | 284.4 (-11.6) | 0.108 | 10.2$\times$ |
> |  | 32 | 25$\rightarrow$10 | 1.97 (+0.19) | 289.5 (-6.5) | 0.105 | 10.5$\times$ |
> |  | 32 | 25$\rightarrow$5 | 1.89 (+0.11) | 293.4 (-2.6) | 0.102 | 10.9$\times$ |
> |  | 32 | 15$\rightarrow$10 | 1.96 (+0.18) | 295.3 (-0.7) | 0.100 | 11.0$\times$ |
> |  | 32 | 15$\rightarrow$5 | 2.11 (+0.33) | 295.1 (-0.9) | 0.097 | 11.4$\times$ |
> | - | - | - | - | - | - | - |
> |  | 16 | 50$\rightarrow$25 | 4.72 (+2.94) | 243.7 (-52.3) | 0.079 | 13.9$\times$ |
> |  | 16 | 50$\rightarrow$15 | 4.52 (+2.74) | 245.4 (-50.6) | 0.074 | 14.9$\times$ |
> |  | 16 | 50$\rightarrow$10 | 4.31 (+2.53) | 247.7 (-48.3) | 0.071 | 15.5$\times$ |
> |  | 16 | 50$\rightarrow$5 | 3.86 (+2.08) | 251.8 (-44.2) | 0.069 | 16.0$\times$ |
> | - | - | - | - | - | - | - |
> |  | 8 | 50$\rightarrow$25 | 16.81 (+15.03) | 164.0 (-132.0) | 0.054 | 20.6$\times$ |
> |  | 8 | 50$\rightarrow$15 | 16.34 (+14.56) | 164.5 (-131.5) | 0.050 | 22.3$\times$ |
> |  | 8 | 50$\rightarrow$10 | 15.51 (+13.73) | 169.1 (-126.9) | 0.046 | 23.7$\times$ |
> |  | 8 | 50$\rightarrow$5 | 13.64 (+11.86) | 181.0 (-115.0) | 0.044 | 25.0$\times$ |

---

> > ### Author Response · Authors · 2025-11-19
> >
> > ### MAR-L + DiSA (cosine)
> > | Model | AR steps | Diff steps  | FID$\downarrow$ | IS$\uparrow$ | Time (s)$\downarrow$ | Speed-Up$\uparrow$ |
> > | -- | -- | -- | -- | -- | -- | -- |
> > | MAR-L | 256 | 100 | 1.78 | 296.0 | 1.102 | 1.0$\times$ |
> > | | 64 | 50 | 1.86 (+0.08) | 294.0 (-2.0) | 0.250 | 4.4$\times$ |
> > | +DiSA | 64 | 50$\rightarrow$25 | 1.84 (+0.06) | 291.8 (-4.2) | 0.228 | 4.8$\times$ |
> > |  | 64 | 50$\rightarrow$15 | 1.82 (+0.04) | 292.8 (-3.2) | 0.220 | 5.0$\times$ |
> > |  | 64 | 50$\rightarrow$10 | 1.77 (-0.01) | 296.6 (+0.6) | 0.216 | 5.1$\times$ |
> > |  | 64 | 50$\rightarrow$5 | 1.80 (+0.02) | 297.3 (+1.3) | 0.212 | 5.2$\times$ |
> > | - | - | - | - | - | - | - |
> > |  | 32 | 50$\rightarrow$25 | 2.17 (+0.39) | 279.8 (-16.2) | 0.125 | 8.8$\times$ |
> > |  | 32 | 50$\rightarrow$15 | 2.08 (+0.30) | 283.8 (-12.2) | 0.119 | 9.2$\times$ |
> > |  | 32 | 50$\rightarrow$10 | 2.01 (+0.23) | 283.7 (-12.3) | 0.116 | 9.5$\times$ |
> > |  | 32 | 50$\rightarrow$5 | 1.89 (+0.11) | 287.6 (-8.4) | 0.113 | 9.8$\times$ |
> > | - | - | - | - | - | - | - |
> > |  | 16 | 50$\rightarrow$25 | 4.65 (+2.87) | 242.5 (-53.5) | 0.076 | 14.4$\times$ |
> > |  | 16 | 50$\rightarrow$15 | 4.45 (+2.67) | 245.3 (-50.7) | 0.071 | 15.6$\times$ |
> > |  | 16 | 50$\rightarrow$10 | 4.04 (+2.26) | 250.3 (-45.7) | 0.068 | 16.1$\times$ |
> > |  | 16 | 50$\rightarrow$5 | 3.34 (+1.56) | 261.9 (-34.1) | 0.065 | 16.8$\times$ |
> > | - | - | - | - | - | - | - |
> > |  | 8 | 50$\rightarrow$25 | 16.80 (+15.02) | 162.8 (-133.2) | 0.052 | 21.3$\times$ |
> > |  | 8 | 50$\rightarrow$15 | 16.05 (+14.27) | 165.9 (-130.1) | 0.047 | 23.5$\times$ |
> > |  | 8 | 50$\rightarrow$10 | 15.06 (+13.28) | 172.0 (-124.0) | 0.044 | 24.8$\times$ |
> > |  | 8 | 50$\rightarrow$5 | 12.63 (+10.85) | 186.1 (-109.9) | 0.042 | 26.2$\times$ |
> >
> > ### MAR-L + DiSA (two-stage)
> > | Model | AR steps | Diff steps  | FID$\downarrow$ | IS$\uparrow$ | Time (s)$\downarrow$ | Speed-Up$\uparrow$ |
> > | -- | -- | -- | -- | -- | -- | -- |
> > | MAR-L | 256 | 100 | 1.78 | 296.0 | 1.102 | 1.0$\times$ |
> > | | 64 | 50 | 1.86 (+0.08) | 294.0 (-2.0) | 0.250 | 4.4$\times$ |
> > | +DiSA | 64 | 50$\rightarrow$25 | 1.82 (+0.04) | 292.0 (-4.0) | 0.231 | 4.8$\times$ |
> > |  | 64 | 50$\rightarrow$15 | 1.77 (-0.01) | 296.3 (+0.3) | 0.221 | 5.0$\times$ |
> > |  | 64 | 50$\rightarrow$10 | 1.84 (+0.06) | 294.9 (-1.1) | 0.217 | 5.1$\times$ |
> > |  | 64 | 50$\rightarrow$5 | 2.52 (+0.74) | 290.2 (-5.8) | 0.213 | 5.2$\times$ |
> > | - | - | - | - | - | - | - |
> > |  | 32 | 50$\rightarrow$25 | 2.14 (+0.36) | 282.7 (-13.3) | 0.126 | 8.8$\times$ |
> > |  | 32 | 50$\rightarrow$15 | 2.02 (+0.24) | 282.6 (-13.4) | 0.120 | 9.2$\times$ |
> > |  | 32 | 50$\rightarrow$10 | 1.89 (+0.11) | 288.2 (-7.8) | 0.116 | 9.5$\times$ |
> > |  | 32 | 50$\rightarrow$5 | 2.29 (+0.51) | 290.5 (-5.5) | 0.113 | 9.8$\times$ |
> > | - | - | - | - | - | - | - |
> > |  | 16 | 50$\rightarrow$25 | 4.67 (+2.89) | 243.7 (-52.3) | 0.076 | 14.5$\times$ |
> > |  | 16 | 50$\rightarrow$15 | 4.22 (+2.44) | 249.5 (-46.5) | 0.071 | 15.6$\times$ |
> > |  | 16 | 50$\rightarrow$10 | 3.56 (+1.78) | 256.4 (-39.6) | 0.068 | 16.3$\times$ |
> > |  | 16 | 50$\rightarrow$5 | 2.80 (+1.02) | 269.0 (-27.0) | 0.065 | 17.0$\times$ |
> > | - | - | - | - | - | - | - |
> > |  | 8 | 50$\rightarrow$25 | 16.95 (+15.17) | 162.4 (-133.6) | 0.051 | 21.5$\times$ |
> > |  | 8 | 50$\rightarrow$15 | 15.71 (+13.93) | 168.2 (-127.8) | 0.046 | 23.7$\times$ |
> > |  | 8 | 50$\rightarrow$10 | 13.39 (+11.61) | 181.0 (-115.0) | 0.044 | 25.2$\times$ |
> > |  | 8 | 50$\rightarrow$5 | 8.68 (+6.90) | 209.6 (-86.4) | 0.041 | 26.8$\times$ |

---

> > > ### Author Response · Authors · 2025-11-19
> > >
> > > ### FlowAR-H + DiSA (linear)
> > > | Model | AR steps | Diff steps  | FID$\downarrow$ | IS$\uparrow$ | Time (s)$\downarrow$ | Speed-Up$\uparrow$ |
> > > | -- | -- | -- | -- | -- | -- | -- |
> > > | FlowAR-H | 5 | 50 | 1.67 | 276.3 | 0.423 | 1.0$\times$ |
> > > | +DiSA | 5 | 50$\rightarrow$25 | 1.67 (+0.00) | 273.4 (-2.9) | 0.227 | 1.9$\times$ |
> > > |  | 5 | 50$\rightarrow$15 | 1.69 (+0.02) | 273.6 (-2.7) | 0.167 | 2.5$\times$ |
> > > |  | 5 | 50$\rightarrow$10 | 1.75 (+0.08) | 274.1 (-2.2) | 0.149 | 2.8$\times$ |
> > > |  | 5 | 50$\rightarrow$5 | 3.67 (+2.00) | 227.3 (-49.0) | 0.106 | 4.0$\times$ |
> > > | - | - | - | - | - | - | - |
> > > | | 5 | 25 | 1.72 (+0.05) | 280.7 (+4.4) | 0.198 | 2.1$\times$ |
> > > | +DiSA | 5 | 25$\rightarrow$15 | 1.78 (+0.11) | 280.2 (+3.9) | 0.132 | 3.2$\times$ |
> > > |  | 5 | 25$\rightarrow$10 | 1.88 (+0.21) | 279.2 (+2.9) | 0.105 | 4.0$\times$ |
> > > |  | 5 | 25$\rightarrow$5 | 3.80 (+2.13) | 234.3 (-42.0) | 0.073 | 5.8$\times$ |
> > >
> > >
> > > ### FlowAR-H + DiSA (cosine)
> > > | Model | AR steps | Diff steps  | FID$\downarrow$ | IS$\uparrow$ | Time (s)$\downarrow$ | Speed-Up$\uparrow$ |
> > > | -- | -- | -- | -- | -- | -- | -- |
> > > | FlowAR-H | 5 | 50 | 1.67 | 276.3 | 0.423 | 1.0$\times$ |
> > > | +DiSA | 5 | 50$\rightarrow$25 | 1.67 (+0.00) | 272.4 (-3.9) | 0.243 | 1.7$\times$ |
> > > |  | 5 | 50$\rightarrow$15 | 1.69 (+0.02) | 273.2 (-3.1) | 0.175 | 2.4$\times$ |
> > > |  | 5 | 50$\rightarrow$10 | 1.76 (+0.09) | 271.6 (-4.7) | 0.144 | 2.9$\times$ |
> > > |  | 5 | 50$\rightarrow$5 | 3.92 (+2.25) | 224.5 (-51.8) | 0.103 | 4.1$\times$ |
> > > | | 5 | 25 | 1.72 (+0.05) | 280.7 (+4.4) | 0.198 | 2.1$\times$ |
> > > |  | 5 | 25$\rightarrow$15 | 1.77 (+0.10) | 279.7 (+3.4) | 0.150 | 2.8$\times$ |
> > > |  | 5 | 25$\rightarrow$10 | 1.86 (+0.19) | 277.5 (+1.2) | 0.122 | 3.5$\times$ |
> > > |  | 5 | 25$\rightarrow$5 | 4.48 (+2.81) | 223.4 (-52.9) | 0.076 | 5.6$\times$ |
> > >
> > > ### FlowAR-H + DiSA (two-stage)
> > > | Model | AR steps | Diff steps  | FID$\downarrow$ | IS$\uparrow$ | Time (s)$\downarrow$ | Speed-Up$\uparrow$ |
> > > | -- | -- | -- | -- | -- | -- | -- |
> > > | FlowAR-H | 5 | 50 | 1.67 | 276.3 | 0.423 | 1.0$\times$ |
> > > | +DiSA | 5 | 50$\rightarrow$25 | 1.67 (+0.00) | 274.2 (-2.1) | 0.245 | 1.7$\times$ |
> > > |  | 5 | 50$\rightarrow$15 | 1.72 (+0.05) | 273.3 (-3.0) | 0.178 | 2.4$\times$ |
> > > |  | 5 | 50$\rightarrow$10 | 1.83 (+0.16) | 272.6 (-3.7) | 0.149 | 2.8$\times$ |
> > > |  | 5 | 50$\rightarrow$5 | 6.22 (+4.55) | 194.3 (-82.0) | 0.108 | 3.9$\times$ |
> > > | - | - | - | - | - | - | - |
> > > | | 5 | 25 | 1.72 (+0.05) | 280.7 (+4.4) | 0.198 | 2.1$\times$ |
> > > | +DiSA | 5 | 25$\rightarrow$15 | 1.77 (+0.10) | 280.0 (+3.7) | 0.151 | 2.8$\times$ |
> > > |  | 5 | 25$\rightarrow$10 | 1.87 (+0.20) | 279.6 (+3.3) | 0.122 | 3.5$\times$ |
> > > |  | 5 | 25$\rightarrow$5 | 6.04 (+4.37) | 199.8 (-76.5) | 0.078 | 5.4$\times$ |

---

> ### Author Response · Authors · 2025-11-19
>
> ### xAR-H + DiSA (linear)
> | Model | AR steps | Diff steps  | FID$\downarrow$ | IS$\uparrow$ | Time (s)$\downarrow$ | Speed-Up$\uparrow$ |
> | -- | -- | -- | -- | -- | -- | -- |
> | xAR-H | 4 | 50 | 1.24 | 301.6 | 0.896 | 1.0$\times$ |
> | +DiSA | 4 | 50$\rightarrow$25 | 1.24 (+0.00) | 301.0 (-0.6) | 0.669 | 1.3$\times$ |
> |  | 4 | 50$\rightarrow$15 | 1.23 (-0.01) | 300.3 (-1.3) | 0.577 | 1.6$\times$ |
> |  | 4 | 50$\rightarrow$10 | 1.27 (+0.03) | 297.6 (-4.0) | 0.532 | 1.7$\times$ |
> |  | 4 | 50$\rightarrow$5 | 2.94 (+1.70) | 258.3 (-43.3) | 0.491 | 1.8$\times$ |
> | - | - | - | - | - | - | - |
> |  | 4 | 25 | 1.36 (+0.12) | 294.0 (-7.6) | 0.457 | 2.0$\times$ |
> | +DiSA | 4 | 25$\rightarrow$15 | 1.45 (+0.21) | 290.0 (-11.6) | 0.362 | 2.5$\times$ |
> |  | 4 | 25$\rightarrow$10 | 1.68 (+0.44) | 286.6 (-15.0) | 0.321 | 2.8$\times$ |
> |  | 4 | 25$\rightarrow$5 | 4.62 (+3.38) | 236.8 (-64.8) | 0.272 | 3.3$\times$ |
>
> ### xAR-H + DiSA (cosine)
> | Model | AR steps | Diff steps  | FID$\downarrow$ | IS$\uparrow$ | Time (s)$\downarrow$ | Speed-Up$\uparrow$ |
> | -- | -- | -- | -- | -- | -- | -- |
> | xAR-H | 4 | 50 | 1.24 | 301.6 | 0.896 | 1.0$\times$ |
> | +DiSA | 4 | 50$\rightarrow$25 | 1.25 (+0.01) | 299.1 (-2.5) | 0.665 | 1.3$\times$ |
> |  | 4 | 50$\rightarrow$15 | 1.23 (-0.01) | 302.1 (+0.5) | 0.575 | 1.6$\times$ |
> |  | 4 | 50$\rightarrow$10 | 1.31 (+0.07) | 296.6 (-5.0) | 0.539 | 1.7$\times$ |
> |  | 4 | 50$\rightarrow$5 | 2.95 (+1.71) | 257.9 (-43.7) | 0.486 | 1.8$\times$ |
> | - | - | - | - | - | - | - |
> |  | 4 | 25 | 1.36 (+0.12) | 294.0 (-7.6) | 0.457 | 2.0$\times$ |
> | +DiSA | 4 | 25$\rightarrow$15 | 1.45 (+0.21) | 291.1 (-10.5) | 0.362 | 2.5$\times$ |
> |  | 4 | 25$\rightarrow$10 | 1.73 (+0.49) | 283.8 (-17.8) | 0.316 | 2.8$\times$ |
> |  | 4 | 25$\rightarrow$5 | 4.84 (+3.60) | 235.7 (-65.9) | 0.276 | 3.2$\times$ |
>
>
> ### xAR-H + DiSA (two-stage)
> | Model | AR steps | Diff steps  | FID$\downarrow$ | IS$\uparrow$ | Time (s)$\downarrow$ | Speed-Up$\uparrow$ |
> | -- | -- | -- | -- | -- | -- | -- |
> | xAR-H | 4 | 50 | 1.24 | 301.6 | 0.896 | 1.0$\times$ |
> | +DiSA | 4 | 50$\rightarrow$25 | 1.27 (+0.03) | 304.1 (+2.5) | 0.780 | 1.1$\times$ |
> |  | 4 | 50$\rightarrow$15 | 1.26 (+0.02) | 301.8 (+0.2) | 0.733 | 1.2$\times$ |
> |  | 4 | 50$\rightarrow$10 | 1.26 (+0.02) | 299.3 (-2.3) | 0.711 | 1.3$\times$ |
> |  | 4 | 50$\rightarrow$5 | 2.70 (+1.46) | 261.3 (-40.3) | 0.690 | 1.3$\times$ |
> | - | - | - | - | - | - | - |
> |  | 4 | 25 | 1.36 (+0.12) | 294.0 (-7.6) | 0.457 | 2.0$\times$ |
> | +DiSA | 4 | 25$\rightarrow$25 | 1.37 (+0.13) | 292.1 (-9.5) | 0.457 | 2.0$\times$ |
> |  | 4 | 25$\rightarrow$15 | 1.39 (+0.15) | 293.5 (-8.1) | 0.410 | 2.2$\times$ |
> |  | 4 | 25$\rightarrow$10 | 1.50 (+0.26) | 289.4 (-12.2) | 0.388 | 2.3$\times$ |
> |  | 4 | 25$\rightarrow$5 | 3.64 (+2.40) | 248.4 (-53.2) | 0.363 | 2.5$\times$ |
>
> These results are partly presented in Tables 5-14 and partly visualized in Figure 4 in our paper.
>
> As shown, although the performance degrades under extreme schedules (e.g., AR step=8, Diff steps 50$\rightarrow$5 for MAR-B), DiSA performs robustly across a wide range of schedule designs.
>
> We would like to note that tuning DiSA is very simple. In our experiments, we use the CFG values recommended by the original papers and set $T_{early}$ match the original diffusion steps used in each model. We only need to tune $T_{late}$. The way that we use DiSA is similar to tuning the CFG values — set a value for $T_{late}$ and check the generation quality.

---

> > ### Author Response · Authors · 2025-11-19
> >
> > > **Extension to high-resolution generation**
> >
> > We extend DiSA to MAR on ImageNet 512x512 generation task and Harmon on 512x512 text-to-image generation. We also find a new AR-diffusion model, NOVA[1] during rebuttal, and successfully apply DiSA to NOVA on 1024x1024 text-to-image generation. Results are shown as follows.
> >
> > [1] NOVA: Autoregressive Video Generation without Vector Quantization. ICLR 2025.
> >
> > ### MAR ImageNet 512x512
> >
> > | Model | AR steps | Diff steps  | FID$\downarrow$ | IS$\uparrow$ | Time (s)$\downarrow$ | Speed-Up$\uparrow$ |
> > | -- | -- | -- | -- | -- | -- | -- |
> > | MAR | 256 | 100 | 1.73 (+0.00) | 287.9 (+0.0) | 3.623 | 1.0$\times$ |
> > |  | 256 | 100$\rightarrow$50 | 1.78 (+0.05) | 285.0 (-2.9) | 3.229 | 1.1$\times$ |
> > |  | 256 | 100$\rightarrow$25 | 1.76 (+0.03) | 283.0 (-4.9) | 3.069 | 1.2$\times$ |
> > |  | 256 | 100$\rightarrow$15 | 1.80 (+0.07) | 284.9 (-3.0) | 3.013 | 1.2$\times$ |
> > |  | 256 | 100$\rightarrow$10 | 1.77 (+0.04) | 286.5 (-1.4) | 2.989 | 1.2$\times$ |
> > |  | 256 | 100$\rightarrow$5 | 1.78 (+0.05) | 284.4 (-3.5) | 2.923 | 1.2$\times$ |
> > | - | - | - | - | - | - | - |
> > |  | 256 | 50 | 1.77 (+0.04) | 284.6 (-3.3) | 2.900 | 1.2$\times$ |
> > |  | 256 | 50$\rightarrow$25 | 1.84 (+0.11) | 286.3 (-1.6) | 2.741 | 1.3$\times$ |
> > |  | 256 | 50$\rightarrow$15 | 1.79 (+0.06) | 287.8 (-0.1) | 2.675 | 1.4$\times$ |
> > |  | 256 | 50$\rightarrow$10 | 1.85 (+0.12) | 287.1 (-0.8) | 2.651 | 1.4$\times$ |
> > |  | 256 | 50$\rightarrow$5 | 1.94 (+0.21) | 283.0 (-4.9) | 2.619 | 1.4$\times$ |
> > | - | - | - | - | - | - | - |
> > |  | 256 | 25 | 1.87 (+0.14) | 289.4 (+1.5) | 2.594 | 1.4$\times$ |
> > |  | 256 | 25$\rightarrow$15 | 1.97 (+0.24) | 288.3 (+0.4) | 2.546 | 1.4$\times$ |
> > |  | 256 | 25$\rightarrow$10 | 2.04 (+0.31) | 286.5 (-1.4) | 2.499 | 1.4$\times$ |
> > |  | 256 | 25$\rightarrow$5 | 2.37 (+0.64) | 279.4 (-8.5) | 2.474 | 1.5$\times$ |
> > | - | - | - | - | - | - | - |
> > |  | 128 | 100 | 1.74 (+0.01) | 285.0 (-2.9) | 1.817 | 2.0$\times$ |
> > |  | 128 | 100$\rightarrow$50 | 1.76 (+0.03) | 286.8 (-1.1) | 1.637 | 2.2$\times$ |
> > |  | 128 | 100$\rightarrow$25 | 1.77 (+0.04) | 283.4 (-4.5) | 1.563 | 2.3$\times$ |
> > |  | 128 | 100$\rightarrow$15 | 1.76 (+0.03) | 286.5 (-1.4) | 1.521 | 2.4$\times$ |
> > |  | 128 | 100$\rightarrow$10 | 1.74 (+0.01) | 285.8 (-2.1) | 1.500 | 2.4$\times$ |
> > |  | 128 | 100$\rightarrow$5 | 1.76 (+0.03) | 284.3 (-3.6) | 1.505 | 2.4$\times$ |
> > | - | - | - | - | - | - | - |
> > |  | 128 | 50 | 1.80 (+0.07) | 285.0 (-2.9) | 1.487 | 2.4$\times$ |
> > |  | 128 | 50$\rightarrow$25 | 1.82 (+0.09) | 285.7 (-2.2) | 1.407 | 2.6$\times$ |
> > |  | 128 | 50$\rightarrow$15 | 1.82 (+0.09) | 286.3 (-1.6) | 1.370 | 2.6$\times$ |
> > |  | 128 | 50$\rightarrow$10 | 1.85 (+0.12) | 285.6 (-2.3) | 1.346 | 2.7$\times$ |
> > |  | 128 | 50$\rightarrow$5 | 1.90 (+0.17) | 283.8 (-4.1) | 1.333 | 2.7$\times$ |
> > | - | - | - | - | - | - | - |
> > |  | 128 | 25 | 1.92 (+0.19) | 288.5 (+0.6) | 1.318 | 2.7$\times$ |
> > |  | 128 | 25$\rightarrow$15 | 1.95 (+0.22) | 287.1 (-0.8) | 1.283 | 2.8$\times$ |
> > |  | 128 | 25$\rightarrow$10 | 1.99 (+0.26) | 287.7 (-0.2) | 1.275 | 2.8$\times$ |
> > |  | 128 | 25$\rightarrow$5 | 2.31 (+0.58) | 279.3 (-8.6) | 1.254 | 2.9$\times$ |
> > | - | - | - | - | - | - | - |
> > |  | 64 | 100 | 1.92 (+0.19) | 278.3 (-9.6) | 1.029 | 3.5$\times$ |
> > |  | 64 | 100$\rightarrow$50 | 1.92 (+0.19) | 278.2 (-9.7) | 0.907 | 4.0$\times$ |
> > |  | 64 | 100$\rightarrow$25 | 1.90 (+0.17) | 279.2 (-8.7) | 0.840 | 4.3$\times$ |
> > |  | 64 | 100$\rightarrow$15 | 1.87 (+0.14) | 283.0 (-4.9) | 0.826 | 4.4$\times$ |
> > |  | 64 | 100$\rightarrow$10 | 1.90 (+0.17) | 279.8 (-8.1) | 0.807 | 4.5$\times$ |
> > |  | 64 | 100$\rightarrow$5 | 1.89 (+0.16) | 278.4 (-9.5) | 0.795 | 4.6$\times$ |
> > | - | - | - | - | - | - | - |
> > |  | 64 | 50 | 1.97 (+0.24) | 280.4 (-7.5) | 0.824 | 4.4$\times$ |
> > |  | 64 | 50$\rightarrow$25 | 1.95 (+0.22) | 282.7 (-5.2) | 0.757 | 4.8$\times$ |
> > |  | 64 | 50$\rightarrow$15 | 1.93 (+0.20) | 281.4 (-6.5) | 0.733 | 4.9$\times$ |
> > |  | 64 | 50$\rightarrow$10 | 1.92 (+0.19) | 284.5 (-3.4) | 0.719 | 5.0$\times$ |
> > |  | 64 | 50$\rightarrow$5 | 1.89 (+0.16) | 282.8 (-5.1) | 0.710 | 5.1$\times$ |
> > | - | - | - | - | - | - | - |
> > |  | 64 | 25 | 2.05 (+0.32) | 283.6 (-4.3) | 0.709 | 5.1$\times$ |
> > |  | 64 | 25$\rightarrow$15 | 2.02 (+0.29) | 282.7 (-5.2) | 0.686 | 5.3$\times$ |
> > |  | 64 | 25$\rightarrow$10 | 2.00 (+0.27) | 283.3 (-4.6) | 0.674 | 5.4$\times$ |
> > |  | 64 | 25$\rightarrow$5 | 2.27 (+0.54) | 279.6 (-8.3) | 0.662 | 5.5$\times$ |

---

> ### Author Response · Authors · 2025-11-19
>
> ### Harmon Text-to-image 512x512
>
> | AR steps | Diff steps    | Single Obj. | Two Obj. | Counting | Colors | Position | Color Attri. | Overall | Time per image (s) |
> | - | - | - | - | - | - | - | - | - | - |
> | 64      | 100            | 0.99        | 0.86     | 0.69     | 0.89   | 0.48     | 0.50          | 0.73    | 40                  |
> | +DiSA | 100$\rightarrow$5          | 0.99        | 0.90     | 0.68     | 0.86   | 0.49     | 0.51          | 0.74    | 24                  |
> |         | 50             | 0.99        | 0.88     | 0.71     | 0.88   | 0.48     | 0.53          | 0.74    | 24                  |
> | +DiSA | 50$\rightarrow$5           | 0.99        | 0.89     | 0.68     | 0.87   | 0.41     | 0.55          | 0.73    | 17                  |
> | 32      | 50             | 0.99        | 0.86     | 0.64     | 0.87   | 0.43     | 0.49          | 0.71    | 12                  |
> | +DiSA | 50$\rightarrow$5           | 0.99        | 0.85     | 0.69     | 0.86   | 0.48     | 0.52          | 0.73    | 8                   |
>
>
> ### NOVA Text-to-image 1024x1024
>
> | AR steps | Diff steps    | Single Obj. | Two Obj. | Counting | Colors | Position | Color Attri. | Overall | Time per image (s) |
> | - | - | - | - | - | - | - | - | - | - |
> | 128 | 25 | 0.99 | 0.86 | 0.54 | 0.87 | 0.33 | 0.58 | 0.69 | 10.5 |
> | +DiSA | 25$\rightarrow$20 | 0.99 | 0.86 | 0.56 | 0.84 | 0.32 | 0.55 | 0.69 | 9.5 |
> |  | 25$\rightarrow$15 | 1.00 | 0.86 | 0.59 | 0.84 | 0.33 | 0.54 | 0.69 | 8.8 |
> |  | 25$\rightarrow$10 | 0.99 | 0.88 | 0.54 | 0.86 | 0.30 | 0.56 | 0.69 | 8.2 |
> |  | 25$\rightarrow$5 | 0.99 | 0.88 | 0.58 | 0.86 | 0.32 | 0.54 | 0.69 | 7.4 |
>
> As shown in our results, DiSA can scale to higher resolutions such as 512×512 and 1024×1024, achieving great acceleration while preserving generation quality. We note that high-resolution tasks involve significantly more tokens than 256×256 generation (e.g., 1024 tokens for 512×512 vs. 256 tokens for 256×256).

---

### Author Response · Authors · 2025-12-04
**Summary of the rebuttal**

We sincerely appreciate the reviewers and the ACs for their efforts in reviewing our paper and their constructive feedback.

---
### Strengths

We are glad that the reviewers acknowledge our strengths, including:

**1. Theoretically supported.** Denoising-path straightness analysis theoretically supports our method, "linking DiSA to the geometry of diffusion ODEs and providing a principled foundation rather than a heuristic adjustment." @Reviewer 9xYm

**2. Simple and useful.** DiSA significantly accelerates AR-diffusion models, "demonstrating elegance through minimal intervention." @Reviewer 9xYm, @Reviewer XbsH, @Reviewer phup

**3. Comprehensive experiments.** Our experiments cover "various image‑level metrics, per‑image time, and complements existing diffusion accelerators", "distinguishing DiSA from earlier works that rely solely on output metrics like FID or IS." @Reviewer 9xYm, @Reviewer XbsH

---
### Weaknesses

Our rebuttal addresses the following concerns through our extensive new experiments and analysis.

**1. Sensitivity of DiSA to hyperparameters.** @Reviewer 9xYm Q1, @Reviewer XbsH Q2, @Reviewer phup Q1

Reviewers wondered how DiSA performs with different hyperparameters. To address this concern, we conduct comprehensive experiments. As shown in the global response, DiSA performs robustly across a wide range of schedule designs, providing great trade-offs between efficiency and quality. Besides, we highlight that DiSA is very simple to tune in practice, similar to adjusting the classifier-free guidance scale.

**2. Extend DiSA to higher resolutions.** @Reviewer 9xYm Q2, @Reviewer XbsH Q5

We extend DiSA to MAR on ImageNet 512x512 generation task, Harmon on 512x512 text-to-image generation, and NOVA on 1024x1024 text-to-image generation. Results in the global response indicate that DiSA can scale well to higher resolutions, achieving great acceleration while preserving generation quality.

**3. Evaluate DiSA with human-aligned metrics.** @Reviewer 9xYm Q3

We apply DiSA to NOVA on the text-to-video generation task. The evaluation is based on VBench, which comprises 16 dimensions with human-aligned metrics. In our experiments, DiSA also works on text-to-video generation with minimal performance degradation.

**4. DiSA variant with heuristics or auto-tuning schedules.** @Reviewer 9xYm Q4, @Reviewer XbsH Q3&Q4

We explore three heuristics to guide the DiSA scheduler, including straightness of denoising paths, variance of diffusion-sampled tokens, and uncertainty in predicting mask tokens. These heuristics can be applied in an offline or online way. Our experiments show that the heuristics lead to comparable performance to our tuned DiSA, which is a promising future direction.

**5. Exploring fine-tuning or joint schedule learning to reduce the train-test gap.** @Reviewer 9xYm Q5

We conduct a preliminary experiment to train MAR with a test-aligned timestep schedule to reduce the train-test gap. The test-aligned MAR converges slightly faster than the original MAR, which is an interesting direction for future work.

**6. Novelty concerns.** @Reviewer XbsH Q1, @Reviewer phup Q2

@Reviewer XbsH thought that our paper lacks “math or intuition.” We respectfully disagree.

**Intuition.** Our key intuition is that, during the AR process, previously generated tokens provide more conditioning. Therefore, later tokens follow more constrained distributions and become easier to generate. We also provide various empirical findings to support this intuition.

**Mathematical support.** We measure the straightness of denoising paths in each AR step, following theories in rectified flow. The stronger straightness in the later AR process indicates that we can use larger step sizes and fewer diffusion steps. This is acknowledged by @Reviewer 9xYm.

@Reviewer phup thought that our method is similar to pure diffusion accelerations such as DDIM and DPM-Solver. We respectfully disagree.

**Motivation.** DDIM builds non-Markovian diffusion processes, and DPM-Solver focuses on reducing numerical errors in ODE-based sampling. In contrast, our motivation comes from a observation unique to AR-diffusion: the conditioning grows stronger along the AR process.

**Mechanism.** Diffusion-only methods reduce the number of diffusion steps uniformly, i.e., the first token and the last token use the same diffusion schedule. DiSA instead performs non-uniform step annealing, using more steps for early tokens and fewer for later ones, which is designed for the AR process.

Moreover, **DiSA is orthogonal to diffusion methods.** In Table 3, combining DiSA with methods like DDIM and DPM-Solver can further accelerate MAR while maintaining competitive generation quality.

---

We sincerely thank the reviewers again for their time and constructive feedback. In the discussion stage, no further questions were raised by the reviewers, and we believe that our rebuttal and additional experimental results sufficiently address all their concerns.

---

### Meta-Review · Area_Chair_ePXE · 2026-01-17

**Summary:**

This paper studies why autoregressive (AR) image generators that use diffusion to sample each token are slow, and argues that later AR steps become “easier” because conditioning grows as more tokens are already generated. It supports this with empirical evidence, and proposes DiSA, a training-free inference-time schedule that uses more diffusion steps early and fewer later.

Reviewer Phup will be lowering their score, rendering the overall rating as 6, 4, 4. AC recommends rejection.

**Reviewer Concerns:**

Addressed well by the rebuttal / discussion:
- Sensitivity to schedule / hyperparameters (T_early, T_late; linear/cosine/two-stage): Authors provide extensive additional experiments and argue DiSA is robust across schedule designs and easy to tune in practice.
- Scalability to higher resolution: Authors add results extending DiSA to higher resolutions (e.g., 512×512 and 1024×1024 settings across models).
- Human-aligned / richer evaluation: Authors report applying DiSA to a text-to-video setting and evaluating with VBench (multi-dimensional, human-aligned metrics), addressing the “FID/IS-only” concern at least partially.
- Heuristics / adaptive scheduling variants: Authors explore several heuristics (path straightness, token variance, uncertainty) and report comparable performance to tuned DiSA, supporting feasibility of more adaptive variants.
- Train–test mismatch / need for fine-tuning: Authors include a preliminary test-aligned schedule training experiment, suggesting the gap can be mitigated

Still outstanding:
- Theory / principled derivation of the optimal schedule: A reviewer concern is that DiSA remains somewhat heuristic and lacks a principled derivation for the optimal per-step allocation; the rebuttal acknowledges the difficulty and positions it as future work rather than fully resolving it.
- Novelty framing vs. “porting diffusion acceleration”: While the rebuttal argues DiSA is orthogonal to DDIM/DPM-Solver-style uniform step reduction and can be combined with them, the novelty concern is not fully eliminated (it becomes more a matter of positioning and clarity).

**Reviewer Scores:**

- Reviewer phup: 6 -> 4, as the reviewer explicitly stated.
- Reviewer XbsH: original 4, may be stay at 4
- Reviewer 9xYm: original 6, may stay at 6 or increase to 7.

---

### Decision · Program_Chairs · 2026-01-26

Reject